# MemEIC: A Step Toward Continual and Compositional Knowledge Editing

**Jin Seong**[1,*], **Jiyun Park**[1‡,2,*], **Wencke Liermann**[1], **Hongseok Choi**[1]
**Yoonji Nam**[1‡,3], **Hyun Kim**[1], **Soojong Lim**[1], **Namhoon Lee**[2,†]

Electronics and Telecommunications Research Institute, Republic of Korea[1]
POSTECH[2], Sungkyunkwan University, Korea[3]
{real_castle, wliermann, hongking9, h.kim, isj}@etri.re.kr
{jiyun.park, namhoon.lee}@postech.ac.kr
seinundzeit@g.skku.edu

## Abstract

The dynamic nature of information necessitates continuously updating large vision-language models (LVLMs). While recent knowledge editing techniques hint at promising directions, they often focus on editing a single modality (vision or language) in isolation. This prevalent practice neglects the inherent multimodality of LVLMs and the continuous nature of knowledge updates, potentially leading to suboptimal editing outcomes when considering the interplay between modalities and the need for ongoing knowledge refinement. To address these limitations, we propose MemEIC, a novel method for Continual and Compositional Knowledge Editing (CCKE) in LVLMs. MemEIC enables compositional editing of both visual and textual knowledge sequentially. Our approach employs a hybrid external-internal editor featuring a dual external memory for cross-modal evidence retrieval and dual LoRA adapters that facilitate disentangled parameter updates for each modality. A key component is a brain-inspired knowledge connector, activated selectively for compositional reasoning, that integrates information across different modalities. Experiments demonstrate that MemEIC significantly improves performance on complex multimodal questions and effectively preserves prior edits, setting a new benchmark for CCKE in LVLMs. Our project is available at https://github.com/MemEIC/MemEIC.

## 1 Introduction

Large language models (LLMs) [1–3] can be knowledge edited—that is, updated to incorporate new or corrected facts—without retraining from scratch [4–6]. This capability is crucial to keep LLMs up-to-date with evolving information or correcting errors on the fly. Large vision–language models (LVLMs) [7–9] combine textual and visual understanding, making knowledge editing even more challenging. These models encode factual knowledge in both their textual and visual pathways, and require mechanisms to update both modalities' knowledge consistently. Efficiently editing LVLMs is crucial for real-world applications, such as swiftly correcting critical visual recognition errors (e.g., misidentifying Donald Trump as Boris Johnson as shown in Fig. 1(a)). However, research in multimodal knowledge editing remains scarce compared to the extensive studies in textual LLMs.

---

[*] Equal Contribution.
[†] Corresponding Author.
[‡] Work partially done during an internship at ETRI.

39th Conference on Neural Information Processing Systems (NeurIPS 2025).

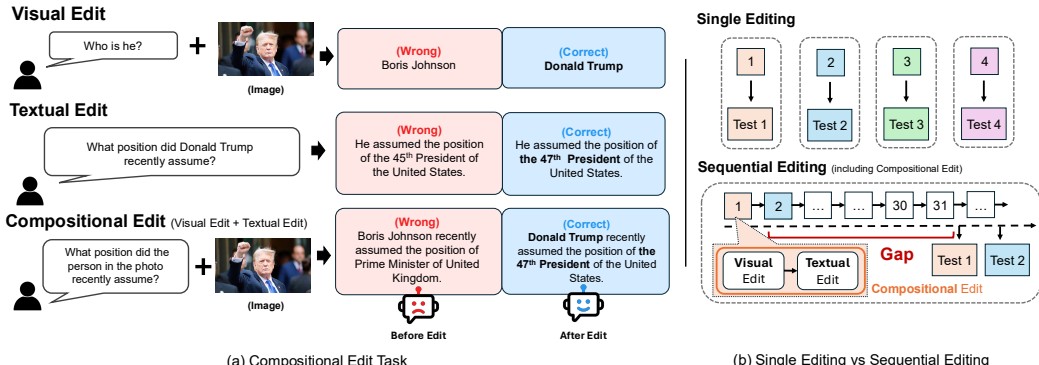

Figure 1: Description of Compositional Edit Task and Sequential Editing

Existing benchmarks for LVLM knowledge editing are limited in scope [10–13]. They typically focus only on visual knowledge updates [10–13], overlooking scenarios where textual facts (e.g., attributes) must also be updated. Furthermore, current benchmarks do not evaluate continual editing [10, 12, 13]—performing multiple sequential knowledge updates—nor do they test a model's ability to handle compositional reasoning [14–16] over edited facts. This is a critical gap because real-world knowledge updates often arrive incrementally and involve intertwined visual and textual information.

To address these shortcomings, we introduce *Continual and Compositional Knowledge Editing* (CCKE), a new benchmark for multimodal knowledge editing. CCKE is the first to assess models under continual knowledge editing [17] and compositional queries that require combining information from both visual and textual edits. We also propose a metric called *Compositional Reliability* (CompRel) to quantify how reliably a model can integrate multiple updated knowledge pieces when answering such complex queries. Evaluating the leading LVLM editors from VLKEB using our new CCKE benchmark (CCKEB) reveals that both external- and internal-memory methods face fundamental challenges in multimodal knowledge editing.

**External memory**-based editor (e.g., SERAC [18], IKE [19]) store edited knowledge externally and retrieve relevant information during inference without modifying model parameters, making them suitable for long-term retention. However, current implementations transplant textual-based retrieval strategies from LLMs directly, ignoring crucial visual cues necessary for accurate multimodal query matching, significantly diminishing retrieval accuracy. Additionally, external memory methods encounter an *internalization* issue, where models overly rely on its stale internal knowledge, especially if the retrieved external information conflicts with what the model has learned, leading to incorrect answers [20–23]. Conversely, **internal memory**-based methods [24–26] embed edits directly into model parameters via fine-tuning or low-rank adaptation (LoRA) [25], viewing the feed-forward network (FFN) itself as a key-value memory structure that inherently stores in-model knowledge [27]. Although beneficial for internalizing edits, these methods typically store visual and textual edits within a unified representation space, resulting in cross-modal interference and representation collapse [28]. Moreover, sequential internal edits suffer from catastrophic forgetting [29–32], degrading previously stored knowledge with successive updates [11].

In this work, we propose **MemEIC**, a novel multimodal knowledge editing framework integrating external retrieval memory with internal model editing. **MemEIC** is designed specifically to handle continual, compositional edits. It operates in several coordinated stages. First, MemEIC uses **query decomposition** to automatically split each user query into its visual part and textual part. Next, for each part, an **external memory** retrieval module fetches relevant information using both image and text cues so that no crucial visual detail or textual context is missed. In parallel, we maintain separate lightweight LoRA adapter modules as **internal memory** for visual and textual knowledge within the model, inspired by *brain lateralization* [33–38]. This knowledge separation ensures that editing a visual fact does not distort textual representations, and vice versa, preventing representation collapse. Crucially, MemEIC introduces a brain-inspired **knowledge connector**—akin to a *corpus callosum* [39, 40]—linking the visual and textual pathways. This connector (using LoRA-augmented self-attention) fuses the edited knowledge from both modalities only when a query requires both to answer. If the query is unimodal, the two streams remain separate. By selectively connecting edited

visual and textual representations, MemEIC enables effective compositional reasoning and yields robust answers even when external retrieval and the model's internal knowledge conflict.

Our contributions are summarized as follows:

- **CCKE Benchmark:** We introduce CCKEB, the first benchmark for continual and compositional multimodal knowledge editing (combining sequential visual and textual edits, with a new reliability metric for evaluation).
- **MemEIC Framework:** We present MemEIC, a multimodal knowledge editing framework that integrates external memory retrieval with internal model editing via modality-specific adapters inspired by brain lateralization, coupled with a brain-inspired knowledge connector to enable robust continual and compositional knowledge reasoning.
- **Empirical Results:** Extensive experiments show that MemEIC achieves interference-free knowledge updates and outperforms prior methods on both edit success and compositional reasoning tasks.

## 2  Problem Formulation and Benchmark Design

### 2.1  Preliminaries: Knowledge Editing for LVLM

**Visual Knowledge Edit**    Visual edits [10, 11] involve updates to the recognized identity label of an image. For an image $i$ once labeled with entity $e$, the task is to change the label to a new entity $e'$:

$$(i, e) \quad \rightarrow \quad (i, e').$$

For example, changing a person's visual identification from *Boris Johnson* to *Donald Trump*. Such edits reflect common real-world updates, such as correction of misidentified individuals or subject identity changes over time.

**Textual Knowledge Edit**    Textual edits focus on changes in factual knowledge associated with specific entities. We modify a 1-hop factual relation $(e, r, o)$ by replacing the original factual object $o$ with a new object $o'$, simulating how factual statements can become outdated:

$$(e, r, o) \quad \rightarrow \quad (e, r, o').$$

For instance, updating facts like *"Donald Trump is the 45th president"* to *"Donald Trump is the 47th president"*. Such edits simulate real-world factual updates (e.g., job titles, affiliations, roles).

**Single Editing vs Sequential Editing**    Most knowledge editing benchmarks assess only single editing, overlooking real-world continual updates and potential forgetting. Sequential editing [11] executes multiple updates sequentially and runs test queries in varying gaps, thereby probing a model's memory retention and robustness over time (see Fig. 1(b)).

### 2.2  Problem Formulation: Continual, Compositional Knowledge Editing (CCKE)

In real-world multimodal applications, entities often undergo interleaved visual and factual updates. We define *Continual, Compositional Knowledge Editing* (CCKE) as the task of having an LVLM accommodate a sequence of alternating visual and textual edits. For simplicity, we limit our experiments to a paired setting where a visual edit is followed directly by its associated textual edit, followed by pairs of unrelated edits:

$$\underbrace{\text{visual edit}}_{(i,e)\rightarrow(i,e')} \quad \rightarrow \quad \underbrace{\text{textual edit}}_{(e',r,o)\rightarrow(e',r,o')} \quad \rightarrow \quad \ldots$$

During the continual editing process, retention of edited knowledge is evaluated at predefined intervals (gaps). To measure a model's ability to integrate multiple edits to answer a single compositional query, we introduce the ***Compositional Reliability*** (CompRel) metric:

$$\text{CompRel} = \mathop{\mathbb{E}}_{\substack{(i_e, vx_e, vy_e) \sim D_{\text{visual edit}} \\ (tx_e, ty_e) \sim D_{\text{textual edit}} \\ (x_c) \sim \mathcal{C}(vx_e, tx_e)}} \Big[ \mathbb{I}\{f(i_e, x_c; \theta') = ty_e\} \Big], \tag{1}$$

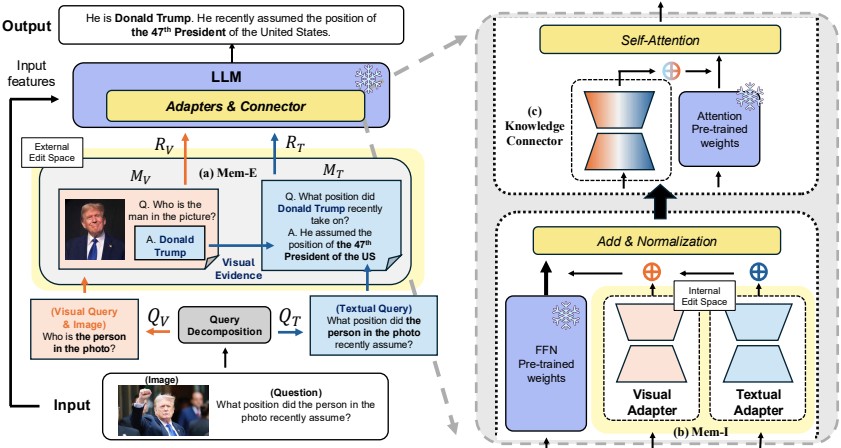

Figure 2: Overall Structure of MemEIC for Continual and Compositional Edit

where $D_{\text{visual edit}}$ denotes the set of visual-edit examples $(i_e, v_{x_e}, v_{y_e})$ with $i_e$ the edited image, $v_{x_e}$ the visual-only query and $v_{y_e}$ its ground-truth answer; $D_{\text{textual edit}}$ denotes the set of textual-edit examples $(t_{x_e}, t_{y_e})$ with $t_{x_e}$ the textual query and $t_{y_e}$ its ground-truth answer; $\theta'$ represent the model parameters after editing and $C$ denotes the operation that composes the visual-edit query $v_{x_e}$, *"Who is in the photo?"*, and the textual-edit query $t_{x_e}$, *"What position did Donald Trump recently assume?"*, into a single compositional query $x_c$, *"What position did the person in the photo recently assume?"*.

## 2.3 Extended Benchmark for Compositional Edit

Our new dataset **CCKEB**, Continual and Compositional Knowledge Editing Benchmark, extends VLKEB by adding *textual* modifications to the existing *visual* edits, thereby forming a fully *compositional* benchmark. Concretely, each image instance in CCKEB is paired with two coordinated edits: (1) a **visual edit** that updates the image's identity label, and (2) a **textual edit** that revises a factual statement about the (new) entity depicted in the image. This design mirrors realistic applications such as correcting outdated factual statements (e.g., *becoming a new president*) while simultaneously fixing mislabeled or changed identities in an image.

**Construction Process**    Following and extending VLKEB [11] construction protocol, we extract knowledge triples and generate textual edits and QA pairs using the multimodal knowledge graph MMKG [41] and GPT-4o [7]. See Appendix B for full details.

## 3 Methodology

In this section, we describe our proposed framework for CCKE. Our approach is designed to address the limitations of existing methods by combining external and internal memory mechanisms while leveraging query decomposition and controlled knowledge integration for improved compositionality. Figure 2 provides an overview of the complete pipeline.

### 3.1 Knowledge Separation

**Query Decomposition.**    To effectively handle compositional queries, we introduce a dedicated module that automatically classifies and decomposes the input textual query into its constituent visual and textual knowledge parts. Formally, given an input query $Q$, the module computes:

$$(Q_v, Q_t) = f(Q)$$

where $Q_v$ and $Q_t$ denote the visual and textual components, respectively, and $f(\cdot)$ represents a query decomposer. This decomposition not only reduces the complexity of the original query but also enables specialized processing pipelines for each modality (Fig. 2(a)). For query decomposition, we used GPT-4o [7]. See Appendix D and E for prompt and decomposition details.

**Modality-Aware External Memory (MEM-E).** In recent LVLM knowledge editing benchmarks [10, 11], external-memory editors directly adopt textual retrieval schemes originally developed for language models, relying solely on textual cues for retrieval and yielding suboptimal visual editing performance. To address this limitation, we propose **Mem-E**, a simple yet effective multimodal external memory storing modality-specific edits in two separate memory storage units as illustrated in Fig. 2(a):

$$M_t = \{(q_i, a_i)\}_{i=1}^{N_t}, \quad M_v = \{(I_j, q_j, a_j)\}_{j=1}^{N_v},$$

where $M_t$ holds textual QA edits and $M_v$ holds visual edits as (image $I$, question $q$, answer $a$) triples. At inference, given a decomposed input textual query $Q_t = Q_t^q$ and visual query $Q_v = (Q_v^I, Q_v^q)$, Mem-E retrieves:

$$i^* = \arg \max_{1 \leq i \leq N_t} \cos\big(\phi_t(Q_t^q), \phi_t(q_i)\big), \qquad\qquad R_t = (q_{i^*}, a_{i^*}),$$

(2)

$$k^* = \arg \max_{1 \leq j \leq N_v} \Big[ \alpha \cos\big(\phi_v(Q_v^I), \phi_v(I_j)\big) + (1-\alpha) \cos\big(\phi_t(Q_v^q), \phi_t(q_j)\big) \Big], \quad R_v = (q_{k^*}, a_{k^*}),$$

(3)

where $\phi_t(\cdot)$ denotes the [CLS] token representation obtained from DistilBERT [42], $\phi_v(\cdot)$ is the CLIP [43] image encoder's [CLS] token representation, and we set $\alpha = 0.5$. For compositional queries composed of textual and visual sub-queries, we first retrieve the target entity $a_k^*$ using $Q_v$. This entity replaces the tagged placeholder in $Q_t^q$, enabling more accurate retrieval from $M_t$.

Finally, for any query $Q$—textual, visual, or compositional, we define the retrieved context

$$R = \begin{cases} R_t, & Q = Q_t, \\ R_v, & Q = Q_v, \\ [R_v; \, R_t], & Q = (Q_v, Q_t), \end{cases}$$

where $[;]$ denotes simple text-level concatenation. This $R$ is then prepended to the original query when prompting the base LVLM, e.g., *"Q: [Image] Who is the man in the picture? A: Donald Trump, Q: What position did Donald Trump recently take on? A: He assumed the position of the $47^{th}$ president of the US. Q: What position did the person in the photo recently assume? A:"*

**Internal Separated Knowledge Integration (MEM-I).** Existing internal memory–based editors view the transformer's feed-forward network (FFN) layers as implicit key–value memories, that can be updated to store new knowledge. Unlike external memory methods, which attach edits externally, these approaches embed all updates in a single shared embedding space. Under continual, compositional knowledge editing, this unified space leads to severe cross-modal interference and catastrophic forgetting as heterogeneous visual and textual updates accumulate.

To overcome these limitations, we propose **Mem-I**, a dual-stream internal memory inspired by human brain lateralization as illustrated in Fig. 2(b). Neuroscience shows that the left hemisphere specializes in language, while the right hemisphere processes visual and spatial information. Analogously, Mem-I extends the FFN memory by two parallel LoRA adapters, connected to the frozen pre-trained FFN weights $\theta$ that retain pre-edit knowledge:

- **Visual adapter** ($\theta_v$) for visual knowledge updates—*right hemisphere*.
- **Textual adapter** ($\theta_t$) for textual knowledge updates—*left hemisphere*.

During the forward pass, we activate modality-specific adapters based on the query type:

- Visual query ($Q_v$): visual adapter $\theta_v$ and frozen FFN $\theta$ are activated.
- Textual query ($Q_t$): textual adapter $\theta_t$ and frozen FFN $\theta$ are activated.
- Compositional query ($(Q_v, Q_t)$): both adapters and frozen FFN $\theta$ are activated.

Analogously, when adding new edits, only the adapter corresponding to the edit modality is updated. As edits are assumed to be atomic, each edit alters either the visual or the textual adapter, never both at once. Concretely, let

$$\theta' = \theta + \begin{cases} \Delta\theta_v, & \text{visual edit,} \\ \Delta\theta_t, & \text{textual edit.} \end{cases}$$

## 3.2  Knowledge Connector

The modality-specific adapters can selectively activate in response to relevant tokens in compositional queries (Appendix E.2). However, when the same sequence contains modality-biased tokens (e.g., one token mainly shaped by the visual adapter, another by the textual), they interact poorly, limiting the model's ability to combine complementary facts during compositional reasoning (Sec. 4.5, *Dual-LoRA + RAG*). To address this, we introduce a simple and effective **Knowledge Connector**, inspired by the corpus callosum in the human brain, which enables information exchange between the two hemispheres. The connector augments the self-attention module with LoRA adapters to facilitate interaction between modality-based token representations (Fig. 2(c)).

Formally, a sequence's hidden state at layer $\ell$ is given as:

$$h^\ell = h_f^{\ell-1} + I_v h_v^{\ell-1} + I_t h_t^{\ell-1}, \tag{4}$$

where $h_f^{\ell-1}$ is the frozen FFN hidden state, $h_v^{\ell-1}$ and $h_t^{\ell-1}$ are the visual and textual adapter states, and $I_v, I_t \in \{0, 1\}$ indicate modality-specific activations. To explicitly model when both adapters are active, we define an indicator function:

$$\mathbb{I}_{v,t} = \begin{cases} 1, & \text{if } I_v = 1 \text{ and } I_t = 1, \\ 0, & \text{otherwise.} \end{cases} \tag{5}$$

Based on the indicator definition above, the Knowledge Connector extends the self-attention module as follows:

$$Q^\ell = h^\ell\big(W_Q + \mathbb{I}_{v,t} \cdot \Delta W_Q^{\mathrm{L}}\big), \quad K^\ell = h^\ell\big(W_K + \mathbb{I}_{v,t} \cdot \Delta W_K^{\mathrm{L}}\big), \quad V^\ell = h^\ell W_V, \tag{6}$$

where $W_Q, W_K, W_V$ are pre-trained projection weights, and $\Delta W_Q^L, \Delta W_K^L$ are LoRA adjustments. In this formulation, the fused hidden state $h^\ell$ provides semantically related token representations retrieved from the frozen FFN, allowing the Knowledge Connector to attend across modality-specific information when both adapters are active. Finally, attention is computed as:

$$z^\ell = \text{Attention}(Q^\ell, K^\ell, V^\ell). \tag{7}$$

This mechanism adaptively integrates information only when both modality-specific adapters are activated, enabling effective interaction between separately maintained visual and textual knowledge (Fig. 4). For unimodal queries, the connector defaults to an identity operation, preserving independent modality representations.

## 4  Experiments

### 4.1  Experimental Setup

Fig. 3 illustrates the overall training and testing setup. Detailed configurations for the experimental setup are provided in Appendix C, and model-specific details in Appendix D.

**Training Stage 1: External Memory**  In the first stage, we freeze the parameters of the base LVLM and train only the external memory module using the CCKEB training set. Each query is decomposed into its visual and/or textual subcomponents through the query decomposition module, depending on the modality required to answer it. The model then learns to retrieve and align each subquery with its corresponding entry in the modality-specific external memory storage, thereby establishing accurate associations between visual and textual evidence.

Table 1: Visual, textual, and compositional reliability evaluated across two backbone models, LLaVA-1.5 and MiniGPT4, on CCKEB testset. See Appendix G for detailed results.

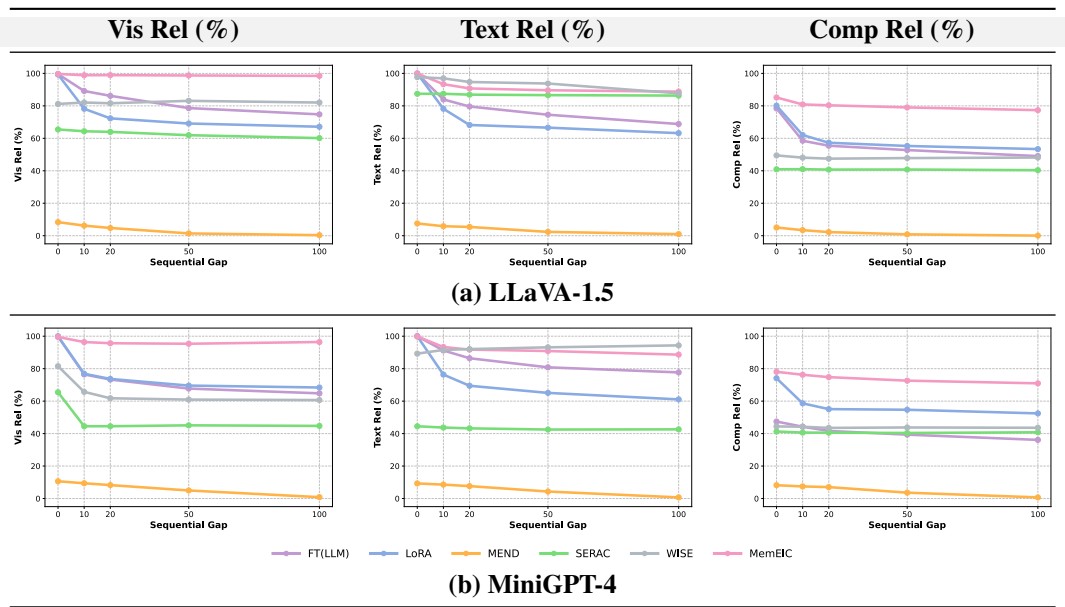

(a) LLaVA-1.5

(b) MiniGPT-4

**Training Stage 2: Knowledge Connector** In the second stage, we activate both the visual and textual adapters and train the Knowledge Connector on the CCKEB training examples using an adversarial retriever. Instead of directly relying on external memory, the retriever provides a mixture of factual and counterfactual evidence to simulate realistic and noisy retrieval conditions. The connector is optimized to fuse the outputs of the two adapters only when a compositional query requires the integration of both visual and textual knowledge, thereby discouraging over-reliance on external memory and encouraging selective integration of internal and external evidence.

**Testing: For Realistic Editing** In deployment, users rarely modify both visual and textual facts simultaneously; rather, edits on different modalities often occur —for example, a textual edit may later be followed by a visual edit as new information becomes available. Hence, we freeze the Knowledge Connector; when an edit targets internal knowledge, we update the corresponding visual or textual adapter, while new evidence is simply appended (stacked) in the external memory store, leaving prior entries and the connector frozen. At test time, we perform 500 editing steps sequentially, evaluating edit retention at gaps of (0, 10, 20, 50, 100).

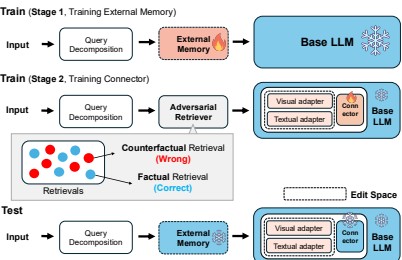

Figure 3: Training Stage and Testing

## 4.2 Main Results

Table 1 demonstrates the competitive performance of the proposed editor, MemEIC, in comparison to four baselines on the CCKEB test set. Baselines include one external memory method, i.e., SERAC [18], and four internal memory editing methods, i.e., FT (LLM), LoRA [25], MEND [24], WISE [44]. Prior work [11] reported that MEND, when applied to sequential editing, caused model collapse and NaN losses. To address this issue, we adopt a greedy interpolation strategy that blends previous weights with newly edited ones. Two baseline candidates, IKE [19] and KE [45], are excluded as they require edit data at test time, which is impractical given evolving knowledge.

**External Memory Methods** Retrieval-based approaches, such as SERAC, store new information externally without modifying any model during edit step, effectively preserving previously learned knowledge. As such, no major performance difference can be observed between the 0 gap and 100

gap setting. However, good reliability depends on correct retrieval. SERAC's text-only retrieval strategy fails to reliably locate visual edits, resulting in lower visual reliability across both backbone models. Additionally, because external methods do not internalize new facts into the model, retrieved facts remain isolated, unable to effectively interact with one another or with the model's internal knowledge, thereby hindering coherent reasoning and leading to diminished compositional reliability.

**Internal Memory Methods**   Fine-tuning or LoRA-based approaches rapidly internalize new knowledge by embedding edited facts into a single shared parameter space, leading to perfect visual and textual reliability in the 0-gap setting. However, when exposed to long sequences of edits, they overwrite previously learned knowledge, causing a drop of nearly 30 points in visual and textual reliability. This is caused by storing heterogeneous visual and textual edits within the same parameters, which leads to representation collapse and severe cross-modal interference. As a result, their compositional reasoning performance degrades substantially as edits accumulate.

To alleviate such forgetting, WISE introduces side-FFN memory slots and a router that interpolates between the original and edited parameters. This design mitigates forgetting and maintains relatively stable performance across sequential gaps. Nevertheless, because edited knowledge is stored as isolated parameter fragments and the router operates primarily on textual activations, WISE remains limited in handling visual edits and fails to fuse visual and textual edits for compositional queries.

**MemEIC—Ours**   MemEIC integrates external and internal memory mechanisms, addressing the limitations of both external-only and internal-only methods. Externally, an image–text retrieval memory reliably stores factual evidence. Internally, lightweight, modality-specific adapters internalize visual and textual edits independently, preventing conflicts with existing knowledge and cross-modal interference. As a result, MemEIC achieves significantly higher performance compared to all baselines and maintains stability across varying edit-test gaps, exhibiting strong robustness against catastrophic forgetting and representation collapse.

Quantitatively, averaged across all edit–test gaps, MemEIC outperforms WISE by +16.94 in visual reliability and by +32.35 in compositional reliability on LLaVA-1.5, demonstrating that temporal robustness alone is insufficient without explicit multimodal fusion. The gap-averaged visual and textual reliability for LLaVA-1.5 reaches 98.93 and 92.48, respectively. This substantial improvement is largely attributed to the brain-inspired *Knowledge Connector*, which dynamically fuses visual and textual representations only when multimodal reasoning is required. With this mechanism, MemEIC achieves an average of 80.56 in compositional reliability, an +18.51 points improvement over the best baseline, LoRA (62.05).

## 4.3   Ablation on External Memory: Visual & Textual Cues in Retrieval

Table 2: Performance Average on VLKEB under Sequential Editing for LLaVA-1.5.

| | Textual | Visual | Add | Rel | T-Gen | I-Gen | T-Loc | I-Loc | Port |
|---|---|---|---|---|---|---|---|---|---|
| SERAC | ✓ | – | ✓ | 75.00 | 45.19 | 75.29 | 99.99 | 1.91 | 38.50 |
| **Mem-E (tex)** | ✓ | – | – | 48.02 | 50.74 | 48.22 | 99.99 | 4.02 | 45.08 |
| **Mem-E (tex+vis)** | ✓ | ✓ | – | **96.51** | **93.42** | **79.90** | 99.99 | **57.10** | **62.09** |

Table 2 compares our external memory-based editing approach (Mem-E) with the SERAC baseline. SERAC [18] relies solely on textual information, i.e., a [CLS] token representation obtained from BERT [46], for retrieving edited knowledge. If a query falls within the scope of an edit, the retrieved information is passed to a separate counterfactual model that "adds" the retrieved information on top of the base model. In contrast, our method uses information retrieved from external memory directly without employing any extra model. We examine two retrieval variants: Mem-E (tex), using textual cues only, and Mem-E (tex+vis), combining textual and visual cues.

As expected, Mem-E (tex) performs worse than SERAC due to the absence of the additional model. However, incorporating visual cues into retrieval (Mem-E (tex+vis)) significantly enhances performance across key metrics. Reliability (Rel) improves markedly from 48.02 to 96.51, indicating that visual features help retrieve edited knowledge more accurately. Text generality (T-Gen) also shows a major gain, increasing from 50.74 to 93.42, demonstrating how visual context can effectively disambiguate queries that are semantically similar. Most notably, image locality (I-Loc) jumps dramatically

from 4.02 to 57.10, underscoring the crucial role of visual retrieval in correctly distinguishing edited from unedited visual information. Detailed results are in Appendix G.2.

## 4.4 Ablation on Internal Memory: Separated Knowledge Integration

Table 3: Knowledge separation (Mem-I) results on CCKEB validation dataset

| | Visual Edit | | | | | Textual Edit | | |
|---|---|---|---|---|---|---|---|---|
| | Rel | T-Gen | I-Gen | T-Loc | I-Loc | Rel | Gen | Loc |
| **Mem-I (Dual-LoRA, r:8 x 2)** | **74.73** | **71.57** | **74.61** | **80.93** | **12.45** | **77.24** | **73.73** | **68.22** |
| **Mem-I (Single-LoRA, r:16)** | 74.60 | 71.50 | 74.49 | 63.16 | 9.59 | 74.54 | 70.49 | 63.16 |
| Difference | -0.13% | -0.07% | -0.12% | -17.77% | -2.86% | -2.70% | -3.24% | -5.06% |
| **Mem-I (Single-LoRA, r:8)** | 74.06 | 71.02 | 74.08 | 60.42 | 7.90 | 72.12 | 68.29 | 60.42 |
| Difference | -0.67% | -0.55% | -0.53% | -20.51% | -4.55% | -5.12% | -5.44% | -7.80% |

*  $p < 0.05$ (paired t-test)

Table 3, we test the hypothesis that mimicking the brain's hemispheric specialization [33–38]—by maintaining separate internal memory spaces for visual and textual information—can substantially improve the precision and overall effectiveness of knowledge editing in LVLMs. To ensure robustness, each experiment is repeated five times and results are averaged. We hold the total parameter budget constant across conditions: Mem-I (Dual-LoRA, $r = 8 + 8$) maintains separate low-rank memories for each modality, whereas Mem-I (Single-LoRA, $r = 16$) uses a shared memory.

In a detailed performance analysis, Mem-I (Dual-LoRA) shows modality-specific advantages. For visual edits, while the mean differences in some metrics (Rel, T-Gen, I-Gen) appear small, a paired t-test confirms these improvements are statistically significant ($p < 0.05$). However, the clearest evidence lies in the much larger gains in localized editing metrics such as T-Loc (+17.77%) and I-Loc (+2.86%), confirming that modality-specific memory separation precisely confines edits to their targets. For textual edits, Dual-LoRA consistently outperforms Single-LoRA across all metrics—Rel (+2.70%), Gen (+3.24%), and Loc (+5.06%). Notably, reducing the rank in Single-LoRA ($r = 8$) markedly degrades performance relative to $r = 16$, showing that our observed improvements are not artifacts of rank selection. Overall, our results suggest that distinct separation of visual and textual memories during CCKE effectively mitigates representation collapse, enabling precise, localized editing of multimodal knowledge. For a detailed breakdown of performance at each individual sequential gap, refer to Appendix E.2

## 4.5 Ablation on Internal Memory: Knowledge Connector

Figure 4 reports that separating knowledge into internal and external memories and then re-combining them through a dedicated *Knowledge Connector* is the key to stable compositional reasoning.

**External retrieval enhances stability, but cannot integrate facts by itself.** Even under the highly favorable assumption of perfect oracle retrieval (Base+RAG), where all relevant edits are correctly fetched, the model's compositional reliability still levels off at just 64.93%. The reason is simple: the freshly retrieved facts sit side-by-side with one another and the model's old, still-stored facts, and the two sets often disagree. With no mechanism to integrate them, this caps performance, showing that perfect retrieval alone is not enough.

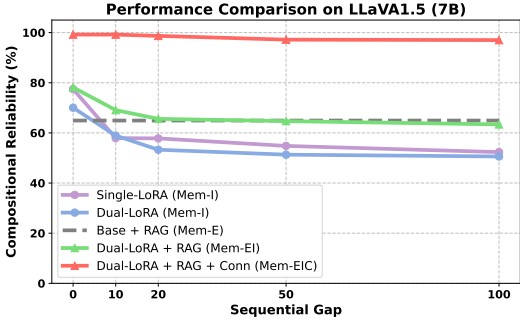

Figure 4: Compositional reliability of five editing variants on LLaVA-1.5 (7B) on CCKEB testset.

**Modality–specific adapters integrate facts, but limit interaction between modalities.** When retrieval is combined with modality-separated adapters (Dual-LoRA+RAG), reliability increases to 78.16 % at gap=0. Nevertheless, performance declines to 63.39 % at gap=100. This decline arises from the absence of alignment between the heterogeneous representations produced by the two

adapters. In comparison to Single-LoRA, Dual-LoRA does not offer a consolidated latent space, that would allow for easy interaction between facts across modalities. This is supported by the observation that Dual-LoRA achieves a compositional reliability of 70.05% at `gap=0`, falling short of the 77.47% attained by Single-LoRA.

**Knowledge Connector unifies modalities for near-oracle reliability.** Augmenting Dual-LoRA+RAG with our attention-based Knowledge Connector raises compositional reliability to 99.21 % at `gap=0` and maintains it at 97.01 % at `gap=100`. These results demonstrate that our Connector effectively combines modality-specific adapters, sustaining near-oracle long-term compositional reasoning.

To evaluate CompRel in a controlled setting, we assumed an oracle external memory that always retrieves the correct knowledge. However, in more realistic scenarios—such as when Mem-E retrieves counterfactual knowledge—conflicts can arise between externally retrieved information and internally edited facts. Even in such cases, our model avoids over-relying on incorrect external knowledge and maintains robust performance (Appendix E.3.2). Also, we explored various connector designs and found that inserting LoRA into the attention projection consistently led to the best performance on the CCKEB (Appendix E.3.4). This finding suggests that LoRA within the attention pathways allows the model to compositionally integrate internally edited knowledge from lower FFN layers, thereby strengthening cross-memory reasoning. The results for MiniGPT-4 are provided in E.3.

## 5 Conclusion

In this paper, we propose CCKEB, a novel continual and compositional knowledge editing benchmark designed to evaluate multimodal consistency over extended edits. We also introduce MemEIC, an integrated memory system that leverages external retrieval, modality-specific dual-LoRA adapters, and a brain-inspired Knowledge Connector for robust multimodal editing. Extensive evaluations demonstrate that MemEIC outperforms existing methods by effectively handling continual updates, preserving previous edits, and achieving significant compositional reliability.

## Acknowledgements

This work was supported by the Institute of Information & Communications Technology Planning & Evaluation (IITP) grant funded by the Korea Government (MSIT) (No. RS-2023-00216011, Development of Artificial Complex Intelligence for Conceptually Understanding and Inferring like Human). It was also supported by the IITP grant funded by the Korea Government (MSIT) (No. RS-2024-00338140, Development of learning and utilization technology to reflect sustainability of generative language models and up-to-dateness over time).

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

# Appendix

In the Appendix, we introduce more details along with additional experimental results, discussions, and related works:

- Appendix A: Related Works
- Appendix B: Dataset Details
- Appendix C: Experimental Setup
- Appendix E: Additional Experimental Results
- Appendix F: Limitations and Broader Impacts
- Appendix G: Original Results

## A  Related Works

**Benchmarks for LVLM Knowledge Editing**  Early research on knowledge editing focused on text-based LLMs, aiming to correct factual errors or update information without full retraining [4–6, 47]. To extend knowledge editing into large vision language models (LVLMs), benchmarks such as MMEdit and VLKEB [10, 11] have been introduced to enable multimodal information updates. However, these benchmarks largely target visual-only edits: MMEdit relies on artificial tasks (e.g., editing objects not present in the image) and focuses solely on single-edit scenarios, where the model is reinitialized after each update—an impractical approach for continuous real-world settings. Moreover, VLKEB does not fully support compositional edits that require joint updates of both visual and textual knowledge. This gap highlights the need for a comprehensive framework capable of handling continuous and compositional knowledge updates.

**Knowledge Editing Methods**  A key distinction in knowledge editing is whether updates are stored externally or integrated within a model's parameters. Retrieval-based methods use external memory to preserve long-term edits without altering model parameters but tend to inherit a text-centric bias, limiting the effective use of visual cues [18]. In contrast, internal memory-based methods embed updates directly into the model, enabling flexible inference [24]; however, they suffer catastrophic forgetting and risk representation collapse when visual and textual characteristics are stored together in a single layer [40]. These limitations underscore the need for approaches that support compositional, continuous knowledge editing in realistic settings.

## B  Dataset Details

Our new benchmark CCKEB, an extended version of VLKEB [11]—originally focused on visual editing—by enabling compositional edits that involve both visual and textual modalities. CCKEB augments 5,000 visual editing instances from the VLKEB training split with their corresponding textual edits, resulting in 5,000 visual-textual training pairs. The same procedure is applied to the evaluation split, resulting in 1,278 compositional evaluation pairs.

### B.1  Construction Process

To construct the textual edits, we follow the data construction process of VLKEB by leveraging the multi-modal knowledge graph MMKG [41]. Where necessary, we retrieve new knowledge triples $(e', r, o)$, and then generate new textual edits by modifying the object entity $o \rightarrow o'$ within existing and new triples.

**Knowledge Triples Extraction**  Unlike the VLKEB evaluation set, which includes explicit knowledge triples and associated QA pairs, included to evaluate portability, the 5,000 training instances in VLKEB do not provide annotated knowledge triples $(e', r, o)$. To address this gap, we extract candidate triples for the training set using the DB15K [48] knowledge base, which was also used in constructing the VLKEB evaluation set.

Among the 5,000 training samples, we successfully extract knowledge triples for 4,428 instances, using a total of 187 relation types in DB15K. In cases where multiple candidate triples were identified,

we prioritize more informative and sparse relations (e.g., location, birthPlace, creator) to select the most salient triple. For the remaining 572 instances where relation triples could not be extracted automatically, we manually align entity mentions with DB15K entries when partially matched. If the entity was entirely missing from DB15K, we manually supplement it by referencing Wikipedia.

**Object Pairs Selection** Given the relational triples $(e', r, o)$, we perform object replacement by substituting the original object $o$ with a new object $o'$.

Following the data construction process of VLKEB, we apply maximum weight bipartite matching [49] to identify object pairs $(o, o')$ with the highest similarity, based on shared relations. The goal of this process is to ensure that $o$ and $o'$ belong to the same or similar semantic categories, thereby making the substitution $o \rightarrow o'$ contextually valid. To facilitate this, we extract the set of relations associated with each object using the FB15K [50] and DB15K [48] knowledge graphs, in that order. We then compute the number of shared relations between every possible object pair and use this count as edge weights in a bipartite graph. Maximum weight bipartite matching is subsequently applied to select the most semantically similar object pairs. In cases where suitable pairs could not be retrieved from the dataset, we generated candidate object replacements using GPT-4o.

For the evaluation set, all object pairs $(o, o')$ are manually reviewed to ensure data quality. This includes identifying and correcting cases where $o = o'$, where duplicate answers are generated, or where a relation (e.g., timeZone, birthPlace) should yield a single answer but multiple answers are produced.

**Compositional & Textual Rel, Gen and Loc Data Construction** For *Textual Reliability* and *Textual Generality*, we generate two semantically equivalent questions $q_1$ and $q_2$ using GPT-4o, based on the knowledge triples $(e', r, o)$ and the selected object pairs $(o, o')$.

For *Textual Locality*, since this metric evaluates whether unrelated knowledge remains unchanged before and after the edit, we reuse the original textual locality data from the VLKEB dataset without additional data generation.

For *Compositional Reliability*, the VLKEB evaluation set already includes questions aligned with the knowledge triples $(e', r, o)$. To generate compositional QA pairs for the newly generated triples in the train set, we leverage the extracted knowledge triples and the corresponding object pairs $(o, o')$, to formulate a QA pair using GPT-4o.

## B.2 Prompt Templates

This subsection presents the prompt templates used in this study for data generation, using GPT-4o as the underlying model.

Table 4 presents the prompt template used to generate candidate objects and corresponding QA pairs for *Textual Reliability* and *Textual Generality* in cases where suitable object pairs could not be retrieved during the object pair selection process.

Table 5 illustrates the prompt template used to generate QA pairs for *Textual Reliability* and *Textual Generality*, given a knowledge triple $(e', r, o')$.

Table 6 shows the prompt template designed to generate *Compositional Reliability* QA pairs in the training set.

Table 4: Prompt template used to generate $o'$ via GPT-4o.

**Prompt Template for Generating $o'$**

**System:**

You are a counterfactual entity generator and question creator.

The input triple may contain 'null' in the object position, representing a missing or unmatchable entity.

Your job is to:

1. Replace each 'null' with a new counterfactual entity that fits the subject and relation.

2. The new entity must:
   - Be of the correct semantic type (e.g., person, location, organization)
   - Be factually incorrect (i.e., not real-world truth)
   - Be plausible and coherent in context

3. Generate two paraphrased questions and one answer based on the modified triple.

**Examples:**

Example 1

```
Triple:  ("Amy Irving", "spouse", [null])
Modified:  [null] -> ["Steve Jobs"]
Q1:  Who is Amy Irving married to?
Q2:  Who is the spouse of Amy Irving?
A: Steve Jobs
```

*[6 in-context demonstrations abbreviated]*

**User:**

```
Triple:("Fort Smith, Arkansas", "timeZone", [null])
```

**System:**

```
Modified:  [null] -> ["Pacific Time Zone"]
Q1:  What time zone does Fort Smith, Arkansas observe?
Q2:  In which time zone is Fort Smith, Arkansas located?
A: Pacific Time Zone
```

Table 5: Prompt template used to generate Textual Reliability, Generality QA pairs via GPT-4o.

**Prompt Template for Textual Reliability, Generality QA Generation**

**System:**

You are a question generator specializing in single-hop Counterfactual QA. Users will provide a triple in the format (subject, relation, object). Each triple intentionally contradicts real-world knowledge.

Your task is to generate two rephrased questions and one answer from the triple, while following the instructions below:

**Task Rules**

1. Always treat the input triple as counterfactual, i.e., it intentionally represents incorrect factual knowledge.
2. Do NOT attempt to correct the triple to match real-world knowledge.
3. Do NOT modify the object if the triple is already:
   - Semantically plausible, Syntactically coherent
   - And type-consistent with the relation.
4. Only modify the object if it is:
   - Semantically incoherent (e.g., wrong type)
   - Syntactically awkward (e.g., meaningless structure)
5. Avoid generating absurd, random, or nonsensical substitutions.
6. The two questions (Q1, Q2) must:
   - Ask about the same information
   - Use different wording or phrasing
   - Not explicitly mention the relation or object
7. The answer (A) must exactly match the object entity.
8. If you modify the object, indicate the change with `Modified:[original_object]->[new_object]`.

**Valid Counterfactual Examples (No Modification Required)**
```
Triple:  ("Barack Obama", "jobTitle", ["Astronaut"])
Q1:  What is Barack Obama's profession?
Q2:  What job does Barack Obama hold?
A: Astronaut
```
*[3 in-context demonstrations abbreviated]*

**Unnatural Triple Examples (Require Modification)**
```
Triple:  ("Eiffel Tower", "location", ["Albert Einstein"])
Modified:  ["Albert Einstein"] -> ["Japan"]
Q1:  Where is the Eiffel Tower located?
Q2:  In which country is the Eiffel Tower situated?
A: Japan
```
*[2 in-context demonstrations abbreviated]*

**User:**

```
Triple:("Fort Smith, Arkansas", "timeZone", "Pacific Time Zone")
```

**System:**

```
Q1:  What time setting does Fort Smith, Arkansas follow?
Q2:  Which time zone does Fort Smith, Arkansas observe?
A: Pacific Time Zone
```

Table 6: Prompt template used to generate Compositional Reliability QA pairs via GPT-4o.

**Prompt Template for Compositional Reliability QA Generation**

**System:**

You are a multimodal question generator. Your task is to create a new question (Q) and a corresponding answer (A) based on the triple, edit_q, and edit_a provided by the user.

**Important Guidelines:**

1. Your task is to generate Q and A only, using the input triple and edit_q&a as a guide. Do not modify or recreate edit_q or edit_a.

2. Ensure that:

   - The generated question (Q) is contextually relevant to the triple and avoids explicitly including the answer.

   - The generated answer (A) aligns accurately with the triple and edit_a.

3. Always output in this exact format:

  `Q: [Generated question]`

  `A: [Generated answer]`

**Examples:**

Example 1:

```
Triple:  ("Fort Smith, Arkansas", "timeZone", "Central Time Zone")
edit_q:  What time zone is Fort Smith in?
edit_a:  Central Time Zone
```

Output:

```
Q: What is the time zone of the location shown in the picture?
A: Central Time Zone
```

*[3 in-context demonstrations abbreviated]*

**User:**

```
Triple:  ("Fort Smith, Arkansas", "timeZone", "Pacific Time Zone")
edit_q:  In which time zone is Fort Smith, Arkansas located?
edit_a:  Pacific Time Zone
```

**System:**

```
Q: What time zone does the city shown in the image belong to?
A: Pacific Time Zone
```

## B.3  Example of Compositional Editing

Here we present an example table 7 for compositional editing that includes both textual and visual edit, along with evaluation metrics including visual reliability, textual reliability, and compositional reliability.

# C  Experiment Details

## C.1  Metrics

We evaluate knowledge editing methods based on three established criteria from prior work: *Reliability*, *Generality*, and *Locality*.

Let $f$ denote a vision-language model that maps an input $x$, optionally accompanied by an image input $i$, to an output $y$. Here, $\theta$ and $\theta'$ represent the model parameters before and after the editing operation, respectively.

**Reliability**    Reliability measures the extent to which the edited model produces the desired target output $y_e$ instead of the original output $y_o$. It is computed as the average accuracy across all edited instances.

Table 7: An example of Visual, Textual, Compositional Reliability

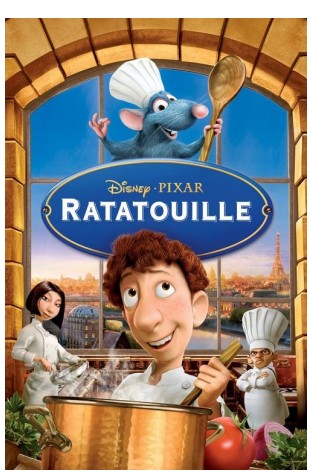

**Visual Edit**
```
Q: What animated film is featured in this image?
Original:  Ratatouille_(film)
Target:  Finding Nemo
```

**Textual Edit**
```
Q: Which company distributed the film Finding Nemo?
Original:  Walt Disney Studios Motion Pictures
Target:  Warner Bros.
```

**Visual Reliability**
```
Q: What animated film is featured in this image?
A: Finding Nemo
```

**Textual Reliability**
```
Q: Which company distributed the film Finding Nemo?
A: Warner Bros.
```

**Compositional Reliability**
```
Q: Which company distributed the movie shown in the picture?
A: Warner Bros.
```

$$\text{VisRel} \;=\; \mathbb{E}_{(i_e,x_e,y_e)\sim\mathcal{D}_{\text{visual edit}}}\Big[\mathbb{I}\{f(i_e,x_e;\theta')=y_e\}\Big], \tag{8}$$

$$\text{TextRel} \;=\; \mathbb{E}_{(x_e,y_e)\sim\mathcal{D}_{\text{textual edit}}}\Big[\mathbb{I}\{f(x_e;\theta')=y_e\}\Big], \tag{9}$$

Here, $\mathcal{D}_{\text{visual edit}}$ and $\mathcal{D}_{\text{textual edit}}$ denotes the dataset used for visual editing and textual editing, respectively.

**Generality**  Generality evaluates whether the edited model can generalize beyond the specific edit case to semantically equivalent inputs across modalities. It is measured by the model's accuracy on perturbed neighbors of the original input, such as rephrased text or modified images.

$$\text{ImageGen} \;=\; \mathbb{E}_{\substack{(i_e,x_e,y_e)\sim\mathcal{D}_{\text{edit}}\\ i_r\sim\mathcal{T}(i_e)}}\Big[\mathbb{I}\{f(i_r,x_e;\theta')=y_e\}\Big], \tag{10}$$

$$\text{TextGen} \;=\; \mathbb{E}_{\substack{([i_e,]x_e,y_e)\sim\mathcal{D}_{\text{edit}}\\ x_r\sim\mathcal{T}(x_e)}}\Big[\mathbb{I}\{f([i_e,]x_r;\theta')=y_e\}\Big], \tag{11}$$

where $\mathcal{T}(x_e)$ and $\mathcal{T}(i_e)$ represent perturbed or semantically modified versions of $x_e$ and $i_e$. Since textual generality can also be evaluated within textual edits, $i_e$ is treated as optional in the *TextGen* setting.

**Locality**   Locality measures the extent to which the edited model preserves its behavior on inputs unrelated to the edit. It is evaluated by comparing the outputs of the original and edited models on out-of-neighbor examples, using both textual (TextLoc) and visual (ImageLoc) probes.

$$\text{ImageLoc} \;=\; \mathbb{E}_{(i_l,x_l,y_l)\sim\mathcal{D}_{\text{loc}}^{\text{image}}}\Big[\mathbb{I}\{f(i_l,x_l;\theta') = f(i_l,x_l;\theta)\}\Big], \tag{12}$$

$$\text{TextLoc} \;=\; \mathbb{E}_{(x_l,y_l)\sim\mathcal{D}_{\text{loc}}^{\text{text}}}\Big[\mathbb{I}\{f(x_l;\theta') = f(x_l;\theta)\}\Big], \tag{13}$$

where $\mathcal{D}_{\text{loc}}^{\text{text}}$ and $\mathcal{D}_{\text{loc}}^{\text{image}}$ represent locality datasets.

## C.2   LVLM Details

The experiments in this study were conducted using two LVLMs: LLaVA-1.5 and MiniGPT4. LLaVA-1.5 employs Vicuna-7B-v1.5 as the language model and ViT-L (0.3B) as the vision encoder, while MiniGPT4 utilizes Vicuna-7B for the language model and ViT-g (1B) for the vision encoder.

## C.3   Knowledge Editing Methods

**FT (Fine-tune)**   In our Fine-tuning (FT) setup, model parameters are updated via gradient descent applied to the designated target layers. Before editing, we store the current weights of these layers to allow restoration after a single-edit operation. We use the AdamW optimizer and configure it such that gradients are computed and applied only to the specified fine-tuning parameters.

**LoRA**   Low-Rank Adaptation (LoRA) [25] provides a parameter-efficient alternative to full fine-tuning by introducing a pair of low-rank trainable matrices into the weight update path of selected layers—typically the query–key–value projections or feed-forward blocks. During training, the original pre-trained weights are frozen, and only the low-rank matrices are optimized, yielding a drastic reduction in the number of trainable parameters and memory footprint. At inference time, the low-rank updates are linearly combined with the frozen base weights, producing an adapted model that preserves the generalization ability of the original network while incorporating task-specific knowledge.

**MEND**   Model Editor Networks with Gradient Decomposition (MEND) [24] offers an intrinsic editing approach that efficiently updates parameters in large language models (LLMs) embedded within vision-language models (LVLMs). Rather than modifying the entire model, MEND trains compact auxiliary networks that apply localized edits based on a single input-output example. These edits are performed by transforming fine-tuning gradients using a low-rank decomposition, enabling scalable and generalizable updates without compromising the model's performance on unrelated tasks.

**SERAC**   Semi-Parametric Editing with a Retrieval-Augmented Counterfactual (SERAC) [18] is a memory-based framework composed of an explicit memory cache—along with an associated scope classifier—and a counterfactual model. The trained scope classifier determines whether an incoming query falls within the editing scope. If the query is in scope, the system forwards the original input together with the retrieved edits from the explicit memory to the counterfactual model; otherwise, the input is passed unchanged to the underlying base model. In our experiments, the scope classifier is instantiated with a BERT encoder(`distilbert-base-cased`), while the counterfactual model varies across LVLMs.

**WISE**   WISE [44] introduces a dual parametric memory for lifelong editing: a main memory that preserves pretrained weights and a side memory (a copied FFN value matrix) that stores edits. A routing activation module decides at inference which memory to use, enabling edited knowledge to be applied only when in scope and preventing collateral changes elsewhere. For continual streams, WISE performs knowledge sharding where different sets of edits reside in distinct and orthogonal subspaces of parameters. Then it merges them to integrate edits without destructive interference; it can also maintain multiple side memories and retrieve by activation (WISE-Retrieve). This routing–sharding–merging design balances reliability, locality, and generalization, overcoming the trade-offs of purely parametric or retrieval-based editors.

## C.4 Experiment Resources and Parameters

All experiments are performed on an NVIDIA A100 80GB GPU utilizing the PyTorch framework. The parameters used in the experiments are summarized in Table 8, and detailed configuration files are available in the code repository. For detailed parameters and configurations specific to MemEIC, please refer to Section D.3.

Table 8: Hyper-parameter settings for LLaVA-1.5 and MiniGPT4 across editing methods

| Method | Model | Steps/ MaxIter | Edit Layer | Optimizer | LR |
|---|---|---|---|---|---|
| FT-LLM | MiniGPT-4 | 10 | 31$^{st}$ layer of Vicuna-7B | AdamW | 1e-4 |
| | LLaVA-1.5 | 10 | 31$^{st}$ layer of Vicuna-7B-v1.5 | AdamW | 1e-4 |
| LoRA | MiniGPT-4 | 10 | all layer of Vicuna-7B | AdamW | 1e-6 |
| | LLaVA-1.5 | 10 | all layer of Vicuna-7B-v1.5 | AdamW | 1e-6 |
| WISE | MiniGPT-4 | 20 | 27$^{th}$ layer of Vicuna-7B | SGD | 1e-1 |
| | LLaVA-1.5 | 20 | 27$^{th}$ layer of Vicuna-7B-v1.5 | SGD | 1e-1 |
| MEND | MiniGPT-4 | 40000 | layer 29, 30, 31 of Vicuna-7B | Adam | 1e-6 |
| | LLaVA-1.5 | 40000 | layer 29, 30, 31 of Vicuna-7B-v1.5 | Adam | 1e-6 |
| SERAC | MiniGPT-4 | 50000 | 31$^{st}$ layer of Vicuna-7B | Adam | 5e-5 |
| | LLaVA-1.5 | 50000 | 31$^{st}$ layer of Vicuna-7B-v1.5 | Adam | 1e-5 |

# D  Model Details

## D.1  Query Decomposition

We automatically classify and decompose each query into its visual and textual components via query decomposition. To perform this step, we utilize GPT-4o, and Table 9 presents the prompt template used for this process.

To ensure consistent and controlled evaluation across various experimental conditions, we preprocess all train and evaluation queries using the query decomposition module. Specifically, query decomposition is performed in advance, resulting in a static set of decomposed queries. However, in practical deployment scenarios, query decomposition must be performed on-the-fly, requiring a full inference pass through the query decomposer such as GPT-4o for each incoming query.

For the performance evaluation of the Query Decomposition Module, please refer to Section E.1

## D.2  External Memory

In the External Memory module (Mem-E), we used `distilbert-base-cased` as a text classifier to compute textual similarity with the stored text edits, and `CLIP-L/336px` as an image encoder to assess visual similarity.

For a detailed description of the parameters and configurations used in the external module training, refer to Section D.3.

## D.3  Training Stage Details

**Stage 1: External Memory**   In the first training stage, we train the text and image classifiers of the external memory module. To prevent temporal interference during classifier training, this stage is conducted under a single-edit setting rather than a continual-edit scenario. The external retrieval module is trained on a datset of 5,000 examples, and Table 10 summarizes the hyperparameter configurations used for LLaVA-1.5 and MiniGPT4. The training objective for the external component is defined as the average binary cross-entropy loss computed over the training dataset.

**Stage 2: Knowledge Connector**   In the second training stage, the Knowledge Connector—implemented as LoRA increments on the self-attention $q/k$ projections of a dedicated fusion layer—is

Table 9: Prompt template for Query Decomposition

**Prompt Template for Query Decomposition**

**System:**

**Task Definition**

You are a powerful query classifier and generator.

Please decompose the question given by the user into image level and text level subquestions using the following steps.

1. Identify the type of the question (e.g., Who, What, Where, When).

2. Identify the key entity or entities relevant to the question.

3. Generate the image level subquestion by focusing on the visual entity(e.g. city featured in the image, person in the picture) in the question. DON'T CHANGE THE MEANING OF THE ORIGINAL QUESTION.

4. Generate the text level subquestion only if additional non-visual information is needed to fully address the query. If not, return 'None'.

**Examples:**

```
Question:  What city is featured in the image?
Image Level:  What [city is featured in the image]?
Text Level:  None

Question:  What is the time zone of the location shown in the picture?
Image Level:  What's [the location shown in the picture]?
Text Level:  What is the time zone of [the location shown in the picture]?
```

*[20 in-context demonstrations abbreviated]*

**User:**

```
Question:  Which league is associated with the club in the picture?
```

**System:**

```
Image Level:  What [club is in the picture]?
Text Level:  Which league is associated with [the club in the picture]?
```

Table 10: Training configuration for Stage 1

| Models | MaxIter | Optimizer | LR |
|--------|---------|-----------|-----|
| LLaVA-1.5 | 50000 | Adam | 1e-5 |
| MiniGPT-4 | 50000 | Adam | 1e-5 |

trained jointly with the *dual-LoRA* adapters. These adapters, positioned in the feed-forward network (FFN) sub-layers of every transformer block, are simultaneously updated whenever an edit occurs. Although not the primary target of fine-tuning, the adapters serve to maintain a consistent feature space for the connector. Training is conducted on the 500 samples of CCKEB train split containing visual, textual, and compositional edits. At each edit step, the relevant adapter(s) and the connector are co-activated and jointly optimized.

To simulate an adversarial retrieval scenario, external-memory hit rates are capped at $70\%$ for LLaVA-1.5 and $50\%$ for MiniGPT-4. The resulting adapter weights are carried over to Stage 3 so that inference receives feature representations drawn from the same distribution observed during training. For both models, we employ the AdamW optimizer with a learning rate of $1e-6$.

# E  Additional Experiments

## E.1  Results on Query Decomposition

To assess the performance of the Query Decomposition Module, we conducted experiments using a decomposed query set, which was conducted by preprocessing queries from the training and

evaluation sets. The evaluation was performed along two axes: **Decomposition Accuracy** and **Semantic Quality**.

Decomposition Accuracy measures whether compositional queries are correctly decomposed into both image-level and text-level sub-queries, and whether visual-only queries are classified solely into image-level components, without being incorrectly decomposed into image-level and text-level sub-queries. Queries that do not include visual components (images) are classified solely as text-level and are therefore excluded from the evaluation. Semantic Quality evaluates the quality and semantic validity of the generated image-level and text-level sub-queries.

Table 11: Percentage distribution of decomposed query types across training and test sets.

|  | **Image & Text Level** | **Image Level** | **Text Level** |
| --- | --- | --- | --- |
| $Q_c$ (Train) | 96.06% | 3.82% | 0.12% |
| $Q_v$ (Train) | 4.98% | 94.99% | 0.03% |
| $Q_c$ (Test) | 97.81% | 2.03% | 0.16% |
| $Q_v$ (Test) | 3.26% | 96.69% | 0.05% |

**Decomposition Accuracy** Table 11 presents the percentage distribution of decomposed query types for Compositional Queries ($Q_c$) and Visual Queries ($Q_v$). For both training and test sets, over 96% of $Q_c$ instances are successfully decomposed into both image-level and text-level components, indicating reliable modality separation. Similarly, for $Q_v$, more than 94% of the instances are accurately decomposed into image-level components only, demonstrating the robustness of the decomposition strategy for visual-only queries.

Table 12: Average scores of decomposed queries evaluated by BERTScore and sBERT similarity

|  | **BERTScore** | **sBERT** |
| --- | --- | --- |
| $Q_c$ (Train) | 93.68 | 85.69 |
| $Q_v$ (Train) | 98.44 | 96.54 |
| $Q_c$ (Test) | 92.92 | 85.71 |
| $Q_v$ (Test) | 99.06 | 97.98 |

**Semantic Quality** Table 12 reports the results of our semantic quality evaluation, presenting the average BERTScore and SBERT cosine similarity between the reference answers and the generated queries for both compositional queries ($Q_c$) and visual queries ($Q_v$). For reference answers, we adopt the original visual reliability questions for image-level compositional queries, and the original compositional queries for text-level cases. For visual queries, the original visual queries are used as reference at the image level. We compute BERTScore using the `distilbert-base-uncased` model and SBERT similarity using the `all-mpnet-base-v2` model, both based on cosine similarity.

As shown in Table 12, compositional queries ($Q_c$) achieve BERTScores above 92 and sBERT scores around 85 on both training and test sets, indicating that the decomposed queries exhibit a relatively high degree of semantic similarity to the reference answers. Due to their structure involving both visual- and text-level sub-queries, compositional queries introduce more variability in expression, which likely contributes to the larger gap observed between BERTScore and sBERT. Visual queries ($Q_v$) consistently yield higher semantic similarity scores across both metrics and data splits, suggesting that queries decomposed from visual queries are more semantically aligned with the ground-truth answers. Moreover, the semantic similarity scores remain stable between training and test sets, indicating the query decomposition module's generalization capability.

### E.2 Results on Internal Memory: Knowledge Separation

Table 13 provides a detailed breakdown of CCKEB performance across varying sequential gaps in the experiments described in Section 4.4. As the gap between the initial edit and subsequent evaluations grows, the Internal Memory–based approach exhibits a pronounced decline in accuracy—evidence of

Table 13: Knowledge separation (Mem-I) results (averaged over 5 runs) for CCKEB validation set

(a) **Visual Edit**

| Gap | Rel | T-Gen | I-Gen | T-Loc | I-Loc |
|---|---|---|---|---|---|
| **Mem-I (Dual-LoRA, r:8 x 2)** | | | | | |
| 0 | 99.96 | 95.53 | 99.96 | 82.53 | 14.62 |
| 10 | 80.02 | 75.41 | 80.75 | 78.09 | 11.70 |
| 20 | 76.08 | 73.87 | 75.52 | 82.88 | 16.26 |
| 50 | 72.61 | 69.93 | 72.70 | 81.68 | 13.67 |
| 100 | 70.24 | 67.09 | 69.50 | 81.07 | 8.17 |
| **Average** | **74.73** | **71.57** | **74.61** | **80.93** | **12.45** |
| **Mem-I (Single-LoRA, r:16)** | | | | | |
| 0 | 99.52 | 95.41 | 99.50 | 67.27 | 9.15 |
| 10 | 80.69 | 75.71 | 80.87 | 70.17 | 11.54 |
| 20 | 75.88 | 72.56 | 75.75 | 59.83 | 8.65 |
| 50 | 72.15 | 70.14 | 72.14 | 64.64 | 8.59 |
| 100 | 69.68 | 67.61 | 69.20 | 58.00 | 6.81 |
| **Average** | 74.60 | 71.50 | 74.49 | 63.16 | 9.59 |
| **Difference** | **-0.13%** | **-0.07%** | **-0.12%** | **-17.77%** | **-2.86%** |
| **Mem-I (Single-LoRA, r:16)** | | | | | |
| **Average** | 74.06 | 71.02 | 74.08 | 60.42 | 7.90 |
| **Difference** | **-0.67%** | **-0.55%** | **-0.53%** | **-20.51%** | **-4.55%** |

(b) **Textual Edit**

| Gap | Rel | Gen | Loc |
|---|---|---|---|
| **Mem-I (Dual-LoRA, r:8 x 2)** | | | |
| 0 | 99.99 | 99.03 | 69.66 |
| 10 | 87.34 | 82.87 | 68.19 |
| 20 | 84.64 | 80.70 | 68.63 |
| 50 | 74.26 | 70.65 | 73.36 |
| 100 | 62.72 | 60.73 | 62.70 |
| **Average** | **77.24** | **73.73** | **68.22** |
| **Mem-I (Single-LoRA, r:16)** | | | |
| 0 | 99.99 | 99.25 | 67.27 |
| 10 | 88.81 | 83.25 | 70.17 |
| 20 | 74.14 | 69.76 | 59.83 |
| 50 | 71.34 | 68.30 | 64.64 |
| 100 | 63.87 | 60.68 | 58.00 |
| **Average** | 74.54 | 70.49 | 63.16 |
| **Difference** | **-2.70%** | **-3.24%** | **-5.06%** |
| **Mem-I (Single-LoRA, r:8)** | | | |
| **Average** | 72.12 | 68.29 | 60.42 |
| **Difference** | **-5.12%** | **-5.44%** | **-7.80%** |

severe catastrophic forgetting when multiple edits accumulate in a shared parameter space. In contrast, our knowledge separation strategy (Mem-I (Dual-LoRA)) sustains robust internal knowledge editing under identical parameter budgets: both Visual Edit and Textual Edit tasks maintain high reliability, generality, and locality metrics, demonstrating the effectiveness of modality-specific memory division in mitigating forgetting.

**Activation Visualization**   Figure 5 presents a heatmap illustrating the impact of each adapter on layer activations in response to different input queries, visual query ($Q_v$) and textual query ($Q_t$). When a $Q_v$ is provided as input, the activations of the Visual Adapter exhibit a pronounced increase, whereas the Textual Adapter demonstrates stronger activation when a $Q_t$ is supplied. This observation indicates that the model effectively leverages the appropriate adapter depending on the modality of the input, selectively utilizing visual and textual information accordingly.

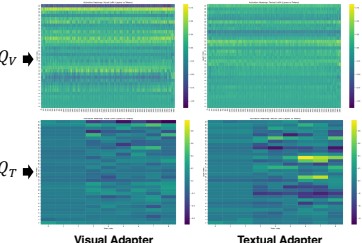

Figure 5: Activation Visualization of Knowledge Separation

### E.3    Additional Results on Knowledge Connector

### E.3.1    Performance Comparison on MiniGPT4

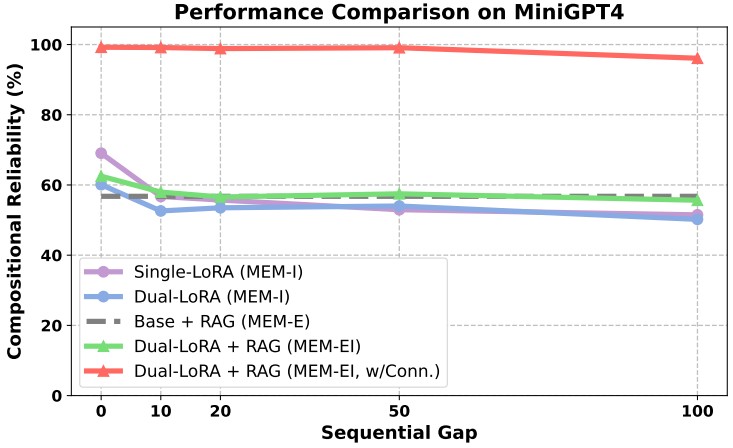

Figure 6: Performance comparison of compositional editing methods methods in Minigpt4.

**Edit configurations**   We group the five systems into three memory regimes:

**Mem-E (Internal Memory)**   LoRA adapters are the *sole* repository of new knowledge.

- Single-LoRA – visual and textual edits share a single feature space.
- Dual-LoRA – visual and textual edits occupy separate feature spaces.

**Mem-I (External Memory)**   No parameters are updated; edited knowledge is retrieved at test time.

- Base + RAG - Minigpt4 with a *oracle* retriever that always returns the correct visual *and* textual memories.

**Mem-EI (External & Internal)**   Internal LoRA adapters are augmented with retrieval.

- Dual-LoRA + RAG – Mem-E + Mem-I, but without an knowledge connector.
- Dual-LoRA + RAG + Connector (**MemEIC, Ours**) – same as above, plus a frozen attention-based Knowledge Connector that fuses the two LoRA streams.

**Results**   Figure 6 shows Compositional Reliability (CompRel) across sequential gaps $g \in \{0, 10, 20, 50, 100\}$.

- **Mem-I (Internal Memory Only)** *Single-LoRA* begins at $69\%$ when $g = 0$, then drops sharply to $56\%$ by $g = 10$ and levels off around $53\%$, indicating severe cross-modal interference. *Dual-LoRA* declines more gradually from $60\%$ to $52\%$, but still cannot maintain high compositional accuracy. This result highlights the inherent limitations of transformer-based internal memory editing, where repeated edits lead to knowledge forgetting [29–31], ultimately degrading compositional reasoning performance.

- **Mem-E (External Memory Only)** Retrieval without any parameter update (*Base+RAG*) remains steady at $56\%$ across all gaps, demonstrating that RAG alone cannot prevent performance decay. This is primarily due to conflicts between externally retrieved knowledge and the model's internal knowledge, causing the baseline model to fail in effective compositional reasoning [20–23].

- **Mem-EI (Internal + External)** Combining Dual-LoRA with retrieval yields a modest gain to $62\%$ at $g = 0$, but this falls to $57\%$ by $g = 10$, reflecting unresolved conflicts between visual and textual edits. Adding the Connector (**MemEIC**) boosts CompRel to $99\%$ at $g = 0$ and sustains $96\%$ at $g = 100$, a consistent $40\%$ improvement over the best non-Connector variant.

### E.3.2 Noisy Retrieval Boosts Compositional Reliability

We train Knowledge Connector variants with fixed *training–time* retrieval accuracies of $50\,\%$, $70\,\%$, and $100\,\%$, and then evaluate each connector under two *test–time* regimes: **Test@50%** (noisy retrieval) and **Test@100%** (oracle retrieval). Figure 7 reports the compositional reliability (CompRel) obtained by the two backbone LVLMs, LLaVA-1.5 and MiniGPT-4.

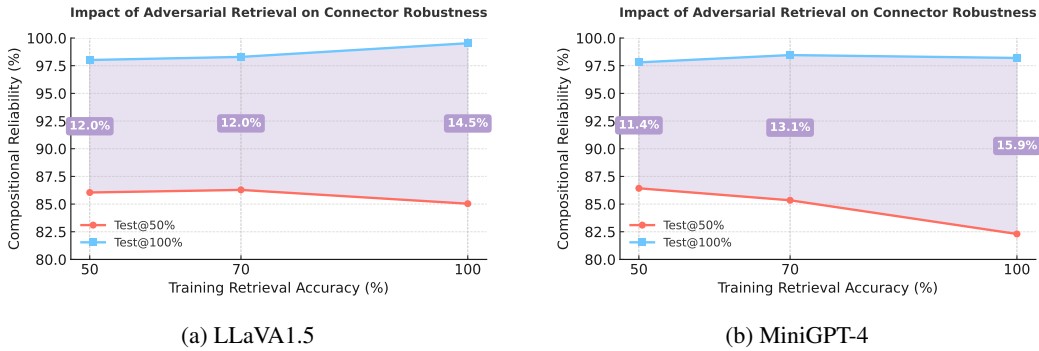

(a) LLaVA1.5            (b) MiniGPT-4

Figure 7: Impact of Adversarial Retrieval on Connector Robustness. Compositional reliability under noisy (Test@50%, red) and perfect (Test@100%, blue) retrieval, after training the knowledge connector with oracle (100%), moderate-noise(70%), or heavy-noise(50%) prompts. Purple shading indicates the robustness gap.

**Results** For **LLaVA-1.5**, the connector trained with *moderate* retrieval noise ($70\,\%$ accuracy) achieves the highest reliability in the noisy setting ($\underline{86.3}\,\%$) while matching the smallest robustness gap ($\underline{12.0}\,\%$). The oracle-trained connector attains the highest ceiling ($\underline{99.5}\,\%$) but suffers the largest gap ($14.5\,\%$).

For **MiniGPT-4** we observe the same qualitative trend, but the optimal trade-off is reached one step earlier, at the *50*%-trained connector: it obtains the best noisy-case reliability ($\underline{86.4}\,\%$) and the narrowest robustness gap ($\underline{11.4}\,\%$). Training with $70\,\%$ accuracy raises the oracle score slightly ($98.4\,\%$) but increases the gap to $13.1\,\%$, while oracle-only training again gives the highest ceiling ($98.0\,\%$) at the cost of the largest gap ($15.9\,\%$).

**Discussion** Across both backbones, introducing a realistic amount of retrieval noise during connector training markedly improves the *robustness–accuracy* balance. With LLaVA-1.5, a 30% error rate (70% training accuracy) is sufficient; the smaller MiniGPT-4 benefits from an even stronger signal, peaking when trained with 50% accuracy. In both cases the connector learns to *cross-check* external evidence against its internal memory: when the retriever is unreliable it falls back on the internal edits,

and when the retriever is correct it yields near-oracle performance. Conversely, training exclusively with perfect retrieval encourages brittle over-reliance on the external prompt, while training with too much noise depresses the attainable ceiling without further narrowing the robustness gap.

These findings confirm that *moderate, task-realistic retrieval noise* is a crucial ingredient for building connectors that remain dependable when deployed in the wild, where external memories are often incomplete or partly incorrect.

### E.3.3    Examples for Connector Robustness

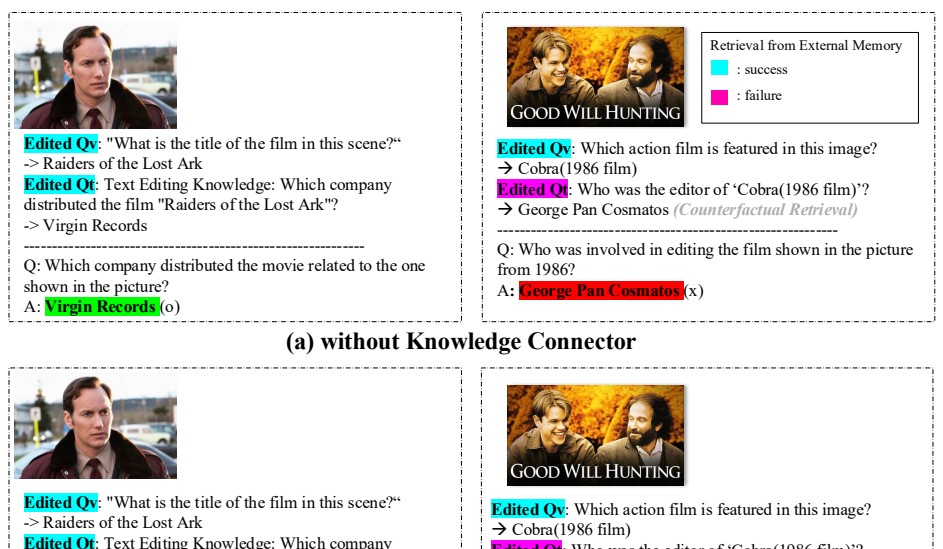

(a) without Knowledge Connector

(b) with Knowledge Connector

Figure 8: Model inference examples with and without the knowledge connector in adversarial retriever setting

Figure 8 presents model inference examples with and without the knowledge connector. In the absence of the knowledge connector, the model produces correct outputs when external memory retrieval is accurate; however, it tends to rely on the retrieved information when the retrieval is incorrect. In contrast, the model with the knowledge connector—trained with an adversarial retriever—leverages internal knowledge alongside the retrieved information. This allows it to generate reliable outputs not only when retrieval is accurate but also when the retrieved information is misleading or incomplete, rather than relying on it solely.

### E.3.4    Additional Results of the Knowledge Connector Selection

We investigate how various configurations of the Knowledge Connector influence the overall performance, as summarized in Table 14, reporting both CompRel and UR for three connector types (MLP, FFN, Attention) under Test@50 retrieval.

To quantify how much the connector improves over internal memory alone, we define the *Knowledge Utilization Ratio(KUR)* as

$$\text{KUR} \;=\; \frac{2 \times \text{CompRel}_{\text{conn}}}{\text{Rel}_{\text{vis}} + \text{Rel}_{\text{tex}}}, \tag{14}$$

which measures the number of correct answers obtained via the connector relative to the average correctness of the visual-only and textual-only internal adapters. A Ratio greater than 1 indicates that the connector yields more correct answers than internal memory alone.

Table 14: Comparison of different connection methods with Test@50% accuracy in CCKEB validation. Ratio indicates the Knowledge Utilization Ratio defined in Eq. 14.

| Method | Conn (MLP), w/ RAG 50%) | | Conn (FFN), w/ RAG 50%) | | Conn (Attention), (w/ RAG 50%) | |
|---|---|---|---|---|---|---|
| | CompRel | Ratio (KUR) | CompRel | Ratio (KUR) | CompRel | Ratio (KUR) |
| Gap 0 | 87.25 | 1.00 | 93.85 | 1.00 | 89.25 | 1.00 |
| 10 | 5.25 | 0.07 | 83.13 | 1.12 | 86.02 | 1.28 |
| 20 | 3.17 | 0.04 | 80.27 | 1.17 | 84.79 | 1.34 |
| 50 | 0.23 | 0.00 | 76.37 | 1.24 | 81.69 | 1.33 |
| 100 | 0.11 | 0.00 | 78.61 | 1.36 | 78.36 | 1.38 |
| **Average** | 19.20 | 0.22 | 82.44 | 1.17 | **84.02** | **1.26** |

**MLP-based Connector**   MLP-based Connector shows strong performance at the initial edit (gap=0: 87.25%) but rapidly loses its connection ability as the gap increases, resulting in an average $CompRel$ of only 19.20% ($KUR = 0.22$). This decline likely stems from its inability to track evolving dual-LoRA features under continual updates, leading to a mismatch between input representations and connector mapping.

**FFN-based Connector**   FFN-based Connector maintains relatively high performance across gaps, achieving an average $CompRel$ of 82.44% ($KUR = 1.17$). However, the performance still drops noticeably as the gap increases, suggesting limited robustness for long-term compositional reasoning despite being a better connector than MLP.

**Attention-based Connector**    implemented by injecting low-rank LoRA increments into the query and key projections of the attention layer, achieves the highest average $CompRel$ of 84.02% ($KUR = 1.26$). By adaptively extracting and fusing modality-specific features from the dual-LoRA modules, the attention-based connector exhibits both higher compositional reliability and better utilization of internal and external knowledge across sequential gaps. Notably, the performance gap at large edit intervals remains minimal compared to FFN and MLP.

In summary, while all connectors play a role in bridging modality-specific memories, the attention-based connector demonstrates superior robustness and knowledge integration, especially under continual, compositional editing scenarios.

# F   Limitations and Broader Impacts

## F.1   Limitations

Despite demonstrating strong empirical results, our approach has several limitations. First, our framework currently focuses solely on multimodal knowledge editing scenarios involving visual recognition and textual understanding, and it does not extend naturally to multimodal generation tasks. Additionally, due to computational resource constraints, MemEIC has primarily been evaluated on moderately sized vision–language models rather than extremely large-scale models; thus, its performance on larger architectures remains unexplored.

## F.2   Broader Impacts

In terms of societal impacts, our research advances the capability of continually updating LVLM models, holding substantial benefits for applications requiring accurate, up-to-date visual and textual knowledge in time-sensitive settings (e.g., misinformation prevention, timely fact correction). However, we acknowledge potential negative implications, such as the risk of malicious actors applying this technology to rapidly propagate disinformation or harmful content. Therefore, we emphasize the importance of deploying rigorous safeguards, monitoring mechanisms, and strong ethical guidelines to ensure responsible use and mitigate the risk of misuse.

# G   Original Results

## G.1   Original Results of Main Results on CCKEB

In Section 4.2, we present the main results of editing performance across varying gap sizes. Tables 15 and 16 report the original results for LLaVA-1.5 and MiniGPT4, respectively.

The first five metrics correspond to those proposed in the original VLKEB benchmark, while the latter four—denoted as *Extended* Metrics—are introduced via the CCKEB benchmark. The VLKEB metrics primarily assess performance on visual edits, whereas the newly constructed T-Rel, T-Gen, and T-Loc metrics evaluate textual edits.

## G.2   Original Results of Ablation Study on External Memory

In Section 4.3, we present the ablation experiment of External Memory. Tables 17 shows the original results for the experiment.

Table 15: Original results in LLaVA-1.5

| Method | Gap | VLKEB | | | | | Extended | | | |
|---|---|---|---|---|---|---|---|---|---|---|
| | | Rel | T-Gen | I-Gen | T-Loc | I-Loc | CompRel | T-Rel | T-Gen | T-Loc |
| FT (LLM) | 0 | 99.57 | 98.19 | 99.05 | 38.09 | 6.66 | 78.42 | 99.87 | 99.12 | 38.09 |
| | 10 | 89.16 | 81.31 | 84.09 | 37.11 | 6.48 | 58.51 | 83.96 | 76.22 | 37.11 |
| | 20 | 86.18 | 77.82 | 80.27 | 36.33 | 6.19 | 55.44 | 79.60 | 71.45 | 36.33 |
| | 50 | 78.63 | 71.86 | 74.09 | 33.76 | 5.41 | 52.76 | 74.54 | 65.14 | 33.76 |
| | 100 | 74.76 | 67.79 | 70.39 | 30.21 | 4.53 | 49.06 | 68.81 | 61.73 | 30.21 |
| LoRA | 0 | 99.32 | 95.77 | 99.33 | 68.34 | 9.56 | 80.14 | 99.92 | 98.96 | 68.34 |
| | 10 | 78.20 | 74.63 | 78.16 | 65.04 | 9.07 | 62.03 | 78.20 | 75.50 | 65.04 |
| | 20 | 72.30 | 70.03 | 72.20 | 62.35 | 8.96 | 57.28 | 68.21 | 64.85 | 62.35 |
| | 50 | 69.12 | 68.77 | 68.89 | 60.06 | 6.82 | 55.29 | 66.58 | 64.92 | 60.06 |
| | 100 | 67.14 | 65.95 | 66.56 | 65.41 | 8.25 | 53.37 | 63.18 | 60.73 | 65.41 |
| MEND | 0 | 8.37 | 7.82 | 8.21 | 15.75 | 5.60 | 5.04 | 7.55 | 7.19 | 15.75 |
| | 10 | 6.22 | 5.31 | 5.89 | 12.64 | 3.48 | 3.43 | 5.85 | 6.11 | 12.64 |
| | 20 | 4.76 | 4.20 | 4.56 | 12.01 | 2.11 | 2.22 | 5.40 | 4.86 | 12.01 |
| | 50 | 1.41 | 1.37 | 1.41 | 6.13 | 0.65 | 0.85 | 2.35 | 2.40 | 6.13 |
| | 100 | 0.33 | 0.57 | 0.30 | 2.09 | 0.08 | 0.06 | 0.98 | 0.50 | 2.09 |
| SERAC | 0 | 65.46 | 42.46 | 65.36 | 100.00 | 1.40 | 40.92 | 87.44 | 56.55 | 100.00 |
| | 10 | 64.35 | 42.28 | 64.25 | 100.00 | 1.40 | 40.99 | 87.44 | 55.70 | 100.00 |
| | 20 | 63.97 | 41.73 | 63.82 | 100.00 | 1.40 | 40.73 | 86.91 | 55.48 | 100.00 |
| | 50 | 61.93 | 41.71 | 61.80 | 100.00 | 1.41 | 40.77 | 86.60 | 54.97 | 100.00 |
| | 100 | 60.13 | 41.28 | 60.12 | 100.00 | 1.41 | 40.38 | 86.33 | 54.75 | 100.00 |
| WISE | 0 | 81.17 | 74.36 | 77.87 | 92.12 | 28.41 | 49.47 | 97.69 | 86.67 | 92.12 |
| | 10 | 82.06 | 72.65 | 76.13 | 92.09 | 27.45 | 48.10 | 96.95 | 80.77 | 92.09 |
| | 20 | 81.60 | 70.27 | 75.05 | 92.02 | 27.19 | 47.47 | 94.70 | 76.95 | 92.02 |
| | 50 | 83.09 | 69.89 | 76.17 | 91.61 | 26.09 | 47.83 | 93.76 | 74.66 | 91.61 |
| | 100 | 82.04 | 69.42 | 74.70 | 91.03 | 25.10 | 48.16 | 87.72 | 70.41 | 91.03 |
| MemEIC | 0 | 99.66 | 98.76 | 93.53 | 100.00 | 47.79 | 85.16 | 99.94 | 99.53 | 100.00 |
| | 10 | 98.86 | 94.60 | 90.71 | 100.00 | 46.90 | 80.88 | 93.36 | 86.05 | 100.00 |
| | 20 | 98.91 | 93.32 | 90.59 | 100.00 | 45.97 | 80.35 | 90.73 | 83.03 | 100.00 |
| | 50 | 98.72 | 92.44 | 89.46 | 100.00 | 43.97 | 79.03 | 89.61 | 77.02 | 100.00 |
| | 100 | 98.48 | 91.74 | 88.06 | 100.00 | 43.17 | 77.36 | 88.77 | 74.40 | 100.00 |

# H   Mathematical Validation of the Knowledge Connector

We now show three simple but important properties of our modality-aware connector, using the notation from Section 3.2.

Table 16: Original results in MiniGPT4

| Method | Gap | VLKEB | | | | | Extended | | | |
|---|---|---|---|---|---|---|---|---|---|---|
| | | **Rel** | **T-Gen** | **I-Gen** | **T-Loc** | **I-Loc** | **CompRel** | **T-Rel** | **T-Gen** | **T-Loc** |
| FT (LLM) | 0 | 99.83 | 99.79 | 99.49 | 44.06 | 4.67 | 47.42 | 99.99 | 97.67 | 44.06 |
| | 10 | 76.46 | 74.74 | 73.69 | 42.87 | 4.30 | 44.24 | 91.20 | 75.41 | 42.87 |
| | 20 | 73.29 | 72.20 | 70.96 | 42.26 | 4.03 | 41.63 | 86.42 | 69.87 | 42.26 |
| | 50 | 67.73 | 66.94 | 66.04 | 40.22 | 3.26 | 39.35 | 80.83 | 64.90 | 40.22 |
| | 100 | 64.80 | 63.62 | 62.59 | 36.05 | 2.54 | 36.12 | 77.70 | 62.46 | 36.05 |
| LoRA | 0 | 99.88 | 96.72 | 99.95 | 66.07 | 14.06 | 74.10 | 99.99 | 99.30 | 66.07 |
| | 10 | 76.87 | 74.30 | 76.63 | 58.14 | 12.09 | 58.64 | 76.35 | 69.91 | 58.14 |
| | 20 | 73.67 | 71.55 | 72.96 | 66.42 | 13.98 | 55.08 | 69.47 | 62.99 | 66.42 |
| | 50 | 69.58 | 68.69 | 69.12 | 62.50 | 12.60 | 54.69 | 65.07 | 63.22 | 62.50 |
| | 100 | 68.40 | 67.29 | 68.37 | 60.08 | 11.32 | 52.44 | 61.07 | 61.02 | 60.08 |
| MEND | 0 | 10.63 | 10.70 | 10.31 | 19.70 | 11.72 | 8.19 | 9.29 | 8.68 | 19.70 |
| | 10 | 9.40 | 9.50 | 9.35 | 17.62 | 9.94 | 7.45 | 8.56 | 8.21 | 17.62 |
| | 20 | 8.26 | 8.42 | 8.32 | 15.84 | 8.28 | 7.03 | 7.64 | 7.26 | 15.84 |
| | 50 | 4.95 | 5.08 | 4.92 | 9.96 | 3.96 | 3.60 | 4.30 | 4.11 | 9.96 |
| | 100 | 0.84 | 0.78 | 0.89 | 1.50 | 0.33 | 0.69 | 0.69 | 0.58 | 1.57 |
| SERAC | 0 | 45.32 | 47.77 | 44.90 | 100.00 | 3.77 | 41.29 | 44.51 | 39.48 | 100.00 |
| | 10 | 44.52 | 47.35 | 44.28 | 100.00 | 3.76 | 40.62 | 43.71 | 39.51 | 100.00 |
| | 20 | 44.53 | 47.28 | 44.19 | 100.00 | 3.78 | 40.56 | 43.23 | 39.51 | 100.00 |
| | 50 | 45.08 | 47.16 | 44.53 | 100.00 | 3.78 | 40.38 | 42.49 | 39.62 | 100.00 |
| | 100 | 44.74 | 46.86 | 44.44 | 100.00 | 3.74 | 40.79 | 42.61 | 38.68 | 100.00 |
| WISE | 0 | 81.52 | 79.81 | 81.87 | 94.36 | 18.52 | 44.42 | 89.27 | 70.55 | 94.36 |
| | 10 | 65.71 | 63.40 | 64.16 | 94.40 | 17.58 | 44.22 | 91.53 | 68.97 | 94.40 |
| | 20 | 61.76 | 61.04 | 60.86 | 94.25 | 17.08 | 43.51 | 92.06 | 67.31 | 94.25 |
| | 50 | 60.94 | 59.65 | 59.34 | 93.98 | 16.47 | 43.77 | 93.15 | 69.62 | 93.98 |
| | 100 | 60.66 | 60.00 | 59.49 | 94.27 | 15.99 | 43.58 | 94.35 | 70.54 | 94.27 |
| MemEIC | 0 | 99.52 | 97.61 | 94.31 | 99.98 | 50.99 | 78.05 | 99.85 | 97.94 | 99.98 |
| | 10 | 96.40 | 88.58 | 87.22 | 99.98 | 50.20 | 76.25 | 93.34 | 85.02 | 99.98 |
| | 20 | 95.67 | 88.07 | 86.31 | 99.98 | 49.41 | 74.78 | 91.74 | 80.93 | 99.98 |
| | 50 | 95.35 | 85.95 | 84.33 | 100.00 | 46.93 | 72.61 | 90.87 | 76.48 | 100.00 |
| | 100 | 96.42 | 85.48 | 85.12 | 100.00 | 45.89 | 70.96 | 88.64 | 73.41 | 100.00 |

Table 17: Experimental results for Sequential Edit in the VLKEB setting using LLaVA-1.5

| Model | Rel | T-Gen | I-Gen | T-Loc | I-Loc | Port |
|---|---|---|---|---|---|---|
| Mem-E (Rv, w/ tex+vis) 0 | 96.51 | 93.42 | 80.10 | 99.43 | 61.57 | 62.09 |
| Mem-E (Rv, w/ tex+vis) 10 | 96.51 | 93.42 | 80.27 | 99.41 | 58.78 | 62.09 |
| Mem-E (Rv, w/ tex+vis) 20 | 96.51 | 93.42 | 80.27 | 99.39 | 56.70 | 62.09 |
| Mem-E (Rv, w/ tex+vis) 50 | 96.51 | 93.42 | 80.27 | 99.41 | 58.78 | 62.09 |
| Mem-E (Rv, w/ tex+vis) 100 | 96.51 | 93.42 | 78.60 | 99.28 | 49.70 | 62.09 |
| Mem-E (Rv, w/ tex) 0 | 48.02 | 50.73 | 48.22 | 100.00 | 4.03 | 45.39 |
| Mem-E (Rv, w/ tex) 10 | 48.02 | 50.73 | 48.22 | 100.00 | 4.03 | 45.21 |
| Mem-E (Rv, w/ tex) 20 | 48.02 | 50.73 | 48.22 | 100.00 | 4.02 | 45.04 |
| Mem-E (Rv, w/ tex) 50 | 48.02 | 50.73 | 48.22 | 100.00 | 4.02 | 44.98 |
| Mem-E (Rv, w/ tex) 100 | 48.02 | 50.81 | 48.22 | 100.00 | 4.02 | 44.78 |
| SERAC 0 | 77.10 | 46.91 | 77.65 | 99.99 | 1.94 | 38.39 |
| SERAC 10 | 76.46 | 46.14 | 77.04 | 99.99 | 1.90 | 38.28 |
| SERAC 20 | 76.12 | 45.69 | 76.24 | 99.99 | 1.90 | 38.95 |
| SERAC 50 | 74.42 | 43.61 | 74.44 | 99.99 | 1.91 | 38.62 |
| SERAC 100 | 70.92 | 43.64 | 71.12 | 99.99 | 1.90 | 38.30 |

### H.1 Commutativity of Adapter Updates

At layer $\ell$, after self-attention and layer-norm we fuse the base FFN and two adapters:

$$h^{(\ell+1)} = h'^{(\ell)} + m_0^{(\ell)} + m_v^{(\ell)} + m_t^{(\ell)}, \tag{B.1}$$

where - $h'^{(\ell)}$ is the hidden state entering the FFN block, - $m_0^{(\ell)}$ is the frozen FFN's output, - $m_v^{(\ell)}, m_t^{(\ell)}$ are the visual/textual LoRA outputs. By basic vector-space algebra,

$$m_v^{(\ell)} + m_t^{(\ell)} = m_t^{(\ell)} + m_v^{(\ell)}, \tag{B.2}$$

so the ordering of modality-specific increments does not change $h^{(\ell+1)}$.

### H.2 Direct-Sum Preservation

Assume the two adapter subspaces intersect only at zero:

$$\mathrm{span}\{m_v^{(\ell)}\} \cap \mathrm{span}\{m_t^{(\ell)}\} = \{0\}. \tag{B.3}$$

Then every combined update

$$x = m_v^{(\ell)} + m_t^{(\ell)} \quad \left(m_v^{(\ell)} \in \mathcal{H}_v,\ m_t^{(\ell)} \in \mathcal{H}_t\right)$$

has a unique decomposition, which by definition means

$$\mathcal{H}_v + \mathcal{H}_t \cong \mathcal{H}_v \oplus \mathcal{H}_t. \tag{B.4}$$

Thus, mixing the two adapters does not lose any representational capacity.

### H.3 Gating Consistency

Recall we gate the connector by

$$\alpha = \mathbf{1}\{I_v = 1 \ \wedge\ I_t = 1\}, \tag{B.5}$$

and define the gated update

$$\tilde{h}^{(\ell+1)} = \alpha\big(h'^{(\ell)} + m_v^{(\ell)} + m_t^{(\ell)}\big) + (1 - \alpha)\big(h'^{(\ell)} + m_0^{(\ell)}\big). \tag{B.6}$$

When $\alpha = 0$, this reduces to

$$\tilde{h}^{(\ell+1)} = h'^{(\ell)} + m_0^{(\ell)}, \tag{B.7}$$

exactly recovering the vanilla FFN branch and ensuring backward compatibility.

