# OpenReview forum: "MemEIC: A Step Toward Continual and Compositional Knowledge Editing"
_NeurIPS.cc/2025/Conference — NeurIPS 2025 poster_

### Official Review · Reviewer_3Fzx · 2025-06-03

**Clarity:** 3
**Significance:** 3
**Originality:** 3
**Rating:** 5
**Confidence:** 4

**Summary:**

The paper addresses the challenge of updating large vision-language models (LVLMs) by proposing a novel method called MemEIC for Continual and Compositional Knowledge Editing (CCKE). MemEIC integrates external memory retrieval with internal model editing, using modality-specific adapters and a brain-inspired knowledge connector to facilitate robust compositional reasoning across visual and textual knowledge. The authors introduce a new benchmark, CCKEB, to evaluate the effectiveness of their approach, demonstrating that MemEIC outperforms existing methods in handling complex multimodal knowledge updates.

**Questions:**

- In table 3, the results are averaged over five times. I am curious about the standard deviations, since the differences are relatively small.
- In the training stage 2: knowledge connector training, what is the effect of the counterfactual retrievals?
- Is it possible to apply the proposed framework on visual-only edit tasks, e.g., MMEdit, or other datasets even if they do not fully support compositional updates? What modification should be taken?

**Ethical Concerns:**

["NO or VERY MINOR ethics concerns only"]

**Final Justification:**

The author cleared all my concerns in the rebuttal. I keep my score.

**Limitations:**

It seems that there is no negative societal impact of this work.
For the limitations, see the weaknesses above.

**Paper Formatting Concerns:**

no concern

**Quality:**

3

**Strengths And Weaknesses:**

strengths

- The paper introduces MemEIC, a novel framework that combines external memory retrieval with internal model editing, addressing the limitations of existing methods that focus on single-modality editing. This dual approach enhances the model's ability to handle complex, compositional queries.
- The creation of the CCKEB benchmark is a significant contribution, as it provides a comprehensive framework for evaluating continual and compositional knowledge editing in LVLMs, filling a gap in existing benchmarks, with the help of multi-modal knowledge graph MMKG and GPT-4o.
- The methodology is well-explained, with clear descriptions of the knowledge separation, external and internal memory mechanisms, and the knowledge connector, facilitating reproducibility and further research.
- The author provides code for reproducing the results in the supplementary material.

weaknesses

- The evaluation is conducted on moderately sized models, and the performance on larger architectures remains unexplored, which could be a limitation for scaling the approach to more complex models.
- Some claims in the paper are ambiguous. See the Questions part.
- Many tables (e.g., Table 3) and plots (e.g., Table 1) lack error bars or confidence intervals, which could affect the interpretation of the results’ statistical significance.

---

> ### Author Rebuttal · Authors · 2025-07-31
>
> We sincerely thank you for recognizing the significance of our work.
>
> We are the first to propose a Continual, Compositional Knowledge Editing Benchmark that takes real-world scenarios into account. As our paper's title suggests, we hope this framework will be "A Step Toward Continual and Compositional Knowledge Editing" and motivate further research in both the multimodal and knowledge editing communities.
>
>
> ---
>
> **Q. Statistical Significance**
> > **W3.** Many tables (e.g., Table 3) and plots (e.g., Table 1) lack error bars or confidence intervals, which could affect the interpretation of the results' statistical significance.
> >
> > **Q1.** In table 3, the results are averaged over five times. I am curious about the standard deviations, since the differences are relatively small.
>
> **A.** Following the reviewer's suggestion, we have updated Table 3 by including the standard deviation (std) and p-values, as shown below.
>
> **Table 3.** Knowledge separation(Mem-I) results for CCKEB validation dataset
> | Model | Rel (Vis) | T‑Gen (Vis) | I‑Gen (Vis) | T‑Loc (Vis) | I‑Loc (Vis) | Rel (Text) | Gen (Text) | Loc (Text) |
> |-|-|-|-|-|-|-|-|-|
> | Mem‑I (Dual‑LoRA, r: 8×2) | 74.73 ± 0.10 | 71.57 ± 0.06 | 74.61 ± 0.08 | 80.93 ± 0.52 | 12.45 ± 0.38 | 77.24 ± 0.10 | 73.73 ± 0.12 | 68.22 ± 0.55 |
> | Mem‑I (Single‑LoRA, r: 16) | 74.60 ± 0.09 | 71.50 ± 0.05 | 74.49 ± 0.07 | 63.16 ± 0.50 | 9.59 ± 0.33 | 74.54 ± 0.11     | 70.49 ± 0.15   | 63.16 ± 0.48   |
> | **Difference (r:16 vs dual)** | –0.13 % | –0.07 %  | –0.12 %  | –17.77 %  | –2.86 %  | –2.70 % | –3.24 % | –5.06 %        |
> | Mem‑I (Single‑LoRA, r: 8)  | 74.06 ± 0.07    | 71.02 ± 0.06   | 74.08 ± 0.07   | 60.42 ± 0.52   | 7.90 ± 0.43 | 72.12 ± 0.07     | 68.29 ± 0.13   | 60.42 ± 0.60   |
> | **Difference (r:8 vs dual)** | –0.67 % | –0.55 % | –0.53 %  | –20.51 % | –4.55 % | –5.12 %  | –5.44 %  | –7.80 %  |
>
>
>
>
> To clarify, the values presented in Table 3 were obtained as follows:
> - For each independent run (we did 5 runs total), performance metrics were measured at sequential gaps (0, 10, 20, 50, 100).
> - In each run, we averaged the five gap‐specific performances to obtain a single gap-averaged value.
> - Across the 5 runs, the standard deviation of those gap-averaged values was ∼ 0.1, indicating low run-to-run variability.
> - Table 3 then reports, for each metric, the mean ± std computed over these five gap-averaged values.
>
> Our additional analysis (moved to §4.3) confirms that, although mean differences between single lora and dual lora in some visual-edit metrics (rel, t-gen, i-gen) are close to one std, paired t-tests (df = 4) yield p < 0.05, indicating these small improvements are statistically significant. The much larger gains in localized editing metrics (T-Loc +17.77 %, I-Loc +2.86 %) remain the clearest evidence that modality-specific memory separation precisely confines edits to their targets.
>
>
> Nevertheless, as explicitly stated in §4.3 of our manuscript, the key message we intend to convey is:
>
> > "Our results suggest that distinct separation of visual and textual memories during CCKE effectively mitigates representation collapse, enabling precise, localized editing of multimodal knowledge."
>
> In fact, substantial performance improvements were observed in other key metrics:
> - Visual edits: Mem-I (Dual-LoRA) achieved significant gains in T-Loc (+17.77 %) and I-Loc (+2.86 %), confirming that modality-specific separation precisely confines edits to targeted knowledge.
> - Textual edits: Mem-I (Dual-LoRA) consistently outperformed Single-LoRA across all metrics (Rel + 2.7 %, Gen + 3.24 %, Loc + 5.06 %).
>
> We will incorporate these additional statistics and explanations into the final manuscript.
>
> ---
>
> **Q. Effect of the counterfactual retrievals**
> > Q2. In the training stage 2: knowledge connector training, what is the effect of the counterfactual retrievals?
>
>
> **A.** During the training stage, the external and internal memory are trained independently from each other, which means that during internal memory training external memory retrievals are simulated.
> It would be possible to simulate 100% correct retrievals, but our experiments have shown that this would inhibit the models capability to handle incorrect retrievals at deployment time.
> In a real-world deployment MemEIC begins by retrieving edited knowledge from the external memory. Any errors at this stage will naturally cascade through the later components.
>
> To assess robustness under such realistic conditions (see Appendix E.3.2), we compared two settings: one in which both training and testing use perfect retrieval (train@100 %/test@100 %) and one in which training remains perfect but testing is noisy (train@100 %/test@50 %). In LLaVA1.5 this mismatch causes a performance drop of approximately 14.5 %.
>
> To mitigate this degradation, we then trained the knowledge connector itself under noisy retrieval conditions (train@50 %) so that it learns to reconcile incomplete or incorrect memory lookups. With this noisy‐training strategy, performance degradation at test@50 % falls from 14.5 % to 12.0 % for LLaVA1.5V (a 2.5 % gain) and from 15.9 % to 11.4 % for MiniGPT-4 (a 4.5 % gain). These improvements reflect the connector's intrinsic knowledge‐reconciliation effects, which we further illustrate with counterfactual retrieval failure examples in Appendix E.3.3.
>
> ---
> **Q. Applicability to Visual-Only Tasks**
> > **Q3.** Is it possible to apply the proposed framework on visual-only edit tasks, e.g., MMEdit, or other datasets even if they do not fully support compositional updates? What modification should be taken?
>
> **A.** Yes, this is possible. Our proposed **MemEIC framework is designed to be modular**, allowing it to be effectively applied to visual-only edit tasks (e.g., MMEdit, VLKEB), simply excluding the knowledge connector.
>
> We will explain this in detail by walking through each operational stage of MemEIC.
>
> **1) Training Stage**
>
> The training of MemEIC consists of two stages.
> - **Stage 1: Training the External Memory (Mem-E):** In this stage, the external memory is trained to retrieve the correct information for a given query. Since MemEIC's external memory is separated into distinct visual and textual units (`M_v` and `M_t`), using a visual-only dataset to train the visual external memory (`M_v`) poses no issue.
> -**Stage 2: Training the Knowledge Connector:** This stage trains the **Knowledge Connector**, which is responsible for fusing information from the visual and textual adapters for compositional reasoning. We train this connector using compositional queries that require both visual and textual knowledge, as illustrated by the prompt structure below.
> ```
> # Retrieved knowledge from external memory
> Visual knowledge: [[Image] <Visual Tokens>, [Text] Q: Who is the person in the photo? A: Donald Trump]
> Textual knowledge: [[Text] Q: What position did Trump recently take on? A: 47th president]
>
> # Query requiring compositional reasoning
> Q: What position did this person in the photo recently assume?
> A: [Answer]
> ```
> However, existing benchmarks like MMEdit and VLKEB lack this type of **compositional (visual+textual) data** by design. Therefore, the Knowledge Connector cannot be effectively trained using these datasets alone.
>
> **2) Edit Stage**
> When a user makes an edit, MemEIC updates two types of memory.
> - **External Memory (Mem-E):** New knowledge is simply appended (stacked) into the external memory store.
> - **Internal Memory (Mem-I):** Only the adapter corresponding to the modality of the edit is selectively updated. For example, a visual edit is performed by solving a VQA task to update only the visual adapter (`θ_v`).
>    * `[[Image] <Visual Tokens>, [Text] Q: Who is the person in the photo?] -> A: Donald Trump`
>
> For a visual-only task, only the visual adapter and visual external memory are updated.
>
> **3) Test (Inference) Stage**
>
> During inference, the Knowledge Connector is selectively activated **only when an input query requires both visual and textual elements**. If a query is unimodal, such as for a visual-only task, the connector remains inactive (defaulting to an identity operation) and does not participate in the inference.
> Since evaluation queries in visual-only edit tasks only require visual information, the Knowledge Connector would not be activated.
>
> **Conclusion.** Based on the analysis above, the necessary modification is very straightforward: **The Knowledge Connector module can simply be excluded.**
> The connector cannot be trained without compositional data, and it remains inactive during inference for visual-only tasks anyway. By removing only the lightweight, LoRA-based Knowledge Connector, the remaining core components of MemEIC (the dual external memory Mem-E and the separated internal adapters Mem-I) can perfectly support visual-only edit tasks. This modularity is a key design feature of our model, allowing for flexible adaptation to various editing scenarios.
>
>
> ---
> **Q. Scaling to Larger Models**
> > **W1.** The evaluation is conducted on moderately sized models, and the performance on larger architectures remains unexplored, which could be a limitation for scaling the approach to more complex models.
>
> **A.** We thank the reviewer for pointing out this important consideration. As noted in Appendix F.1, our current experiments are conducted on mid-sized LVLMs.
>
> However, the key components of MemEIC—dual external memories, modality-specific LoRA adapters, and the Knowledge Connector—are structurally scale-invariant and designed to be modular. In particular, the use of LoRA-based adapters supports efficient scaling, as their parameter- and compute-efficiency becomes increasingly advantageous in larger models.
>
> We plan to explore this direction in future work by extending our evaluation to larger architectures  (e.g., 13B and beyond) to further validate the scalability and robustness of MemEIC across a broader range of model sizes and complexities.

---

> > ### Comment · Reviewer_3Fzx · 2025-08-05
> >
> > Thank you for your response. You have addressed all my concerns.

---

> > > ### Author Response · Authors · 2025-08-06
> > >
> > > Thank you for confirming that our responses fully addressed your concerns. We appreciate your thorough review and will ensure that the discussed clarifications and results are properly reflected in the final manuscript.

---

> > > > ### Author Response · Authors · 2025-08-08
> > > >
> > > > Dear Reviewer 3Fzx,
> > > > Thank you again for your thoughtful feedback throughout the review process.
> > > >
> > > > As outlined in our initial rebuttal, we committed to conducting an additional experiment on scaling to larger models. This experiment has been completed, and the results are presented below for your reference.
> > > >
> > > > To ensure the experiment could be completed within the discussion period, we evaluated on 200 test instances rather than the 500 used in the main results.
> > > >
> > > > **Table 1.** Detailed Performance of CCKEB on LLaVA‑1.5-13B
> > > > | Method | Gap | Vis Rel | Text Rel | Comp Rel |
> > > > |-|-|-|-|-|
> > > > | **FT(LLM)** | 0 | 99.77 | 99.99 | 72.36 |
> > > > | | 10 | 92.06 | 97.58 | 58.82 |
> > > > | | 20 | 89.38 | 97.37 | 54.41 |
> > > > | | 50 | 82.25 | 91.32 | 50.20 |
> > > > | | 100 | 79.08 | 87.02 | 48.21 |
> > > > | **Avg** | | 88.51 | 94.66 | 56.80 |
> > > > | | | | | |
> > > > | **LoRA** | 0 | 98.99 | 99.99 | 83.65 |
> > > > | | 10 | 78.22 | 75.39 | 57.85 |
> > > > | | 20 | 74.20 | 74.05 | 57.74 |
> > > > | | 50 | 69.67 | 65.64 | 55.62 |
> > > > | | 100 | 66.82 | 57.65 | 53.48 |
> > > > | **Avg** | | 77.58 | 74.54 | 61.67 |
> > > > | | | | | |
> > > > | **MemEIC(OURS)** | 0 | 99.40 | 100.00 | 81.23 |
> > > > | | 10 | 97.53 | 97.68 | 79.08 |
> > > > | | 20 | 98.19 | 95.78 | 76.42 |
> > > > | | 50 | 96.95 | 92.82 | 74.11 |
> > > > | | 100 | 98.12 | 91.98 | 72.26 |
> > > > | **Avg** | | **98.04** | **95.65** | **76.62** |
> > > >
> > > > In these results, MemEIC consistently outperforms FT and LoRA across all gap settings.
> > > > As the gap increases (10–100), FT and LoRA experience a steep performance drop in all metrics, whereas MemEIC maintains stable results.
> > > > On average, MemEIC attains 98.04 (Vis), 95.65 (Text), and 76.62 (Comp), substantially higher than both FT and LoRA.
> > > >
> > > > These findings indicate that, even for large-scale models, MemEIC exhibits strong robustness to long-range edits and significantly better compositional reasoning performance compared to baseline methods.
> > > >
> > > > We hope these additional results further clarify the effectiveness of our approach, and we sincerely appreciate your time and consideration during the review process.
> > > >
> > > > Best regards, The Authors

---

### Official Review · Reviewer_qLzo · 2025-07-03

**Clarity:** 2
**Significance:** 2
**Originality:** 2
**Rating:** 4
**Confidence:** 3

**Summary:**

The paper addresses the critical need to continually update large vision-language models (LVLMs), which often struggle to integrate changes across both visual and textual information effectively. Existing methods typically focus on editing just one modality at a time, overlooking the interconnected nature of visual and textual knowledge. To resolve this, the authors propose MemEIC, a novel approach designed specifically for continual and compositional knowledge editing. MemEIC uniquely combines external memory retrieval with internal updates through separate adapters for visual and textual modalities, inspired by the human brain’s functional organization. A central innovation is the brain-inspired "knowledge connector," enabling the model to selectively integrate visual and textual knowledge during compositional reasoning tasks. Additionally, the authors introduce the CCKEB benchmark, a new comprehensive dataset and evaluation metric specifically tailored for assessing models' capabilities in handling continuous and combined visual-textual edits. Experiments confirm that MemEIC significantly outperforms prior methods, demonstrating robustness in complex multimodal scenarios and effectively preserving previously learned knowledge.

**Questions:**

Computational overhead: Could you clarify the practical computational cost of MemEIC? Providing runtime comparisons or complexity analysis versus baseline methods would help.
Robustness to retrieval errors: How does performance degrade when external memory retrieval accuracy is lower? Additional experiments or insights would strengthen confidence in real-world use.
Real-world generalization: Can you discuss or provide examples demonstrating MemEIC’s effectiveness beyond the synthetic benchmark scenarios? Real-world cases or qualitative results would be beneficial.
Simplifying methodological descriptions: Could you offer a simplified, intuitive explanation of the dual adapters and knowledge connector? This would greatly improve clarity.
Qualitative examples: Including more qualitative examples of successful and unsuccessful compositional edits would help clearly illustrate the practical strengths and weaknesses of your approach.

**Ethical Concerns:**

["NO or VERY MINOR ethics concerns only"]

**Limitations:**

The authors have not explicitly discussed limitations and potential negative societal impacts.

**Paper Formatting Concerns:**

No major formatting issues noticed.

**Quality:**

2

**Strengths And Weaknesses:**

Strengths:
* Quality: Extensive experiments clearly show the effectiveness of the proposed MemEIC method against strong baselines. Ablation studies effectively isolate the contributions of key components.
* Clarity: Well-organized and easy to follow. Figures and problem definitions significantly enhance readability.
* Significance: Addresses an important and understudied challenge in multimodal continual knowledge editing. Introducing a new benchmark (CCKEB) and metric (Compositional Reliability) provides valuable tools for future research.
* Originality: Innovatively integrates external memory and modality-specific adapters inspired by human brain structures, which is novel compared to previous single-modality approaches.

Weaknesses:
* Quality: Missing a discussion on computational overhead, which raises concerns about practical scalability.
* Clarity: Methodological sections can sometimes be overly complex; more qualitative examples would help illustrate the practical effectiveness clearly.
* Significance: Evaluations primarily on synthetic benchmarks may limit real-world generalizability. The absence of user studies or real-world validation weakens claims about practicality.
* Originality: Some components (e.g., LoRA adapters, external retrieval) are incremental adaptations, with originality mostly stemming from their novel combination.

Overall, the paper is strong and makes valuable contributions, but addressing practical considerations and incorporating real-world validation would substantially strengthen its impact.

---

> ### Author Rebuttal · Authors · 2025-07-31
>
> We are truly delighted that you recognized the value of our work with the comment:
> > *"Significance: provides valuable tools for future research."*
>
> Our research proposes, for the first time, a benchmark for continual and compositional knowledge editing that considers realistic scenarios. True to the title of our paper, we aimed to create — *“A Step Toward”* — that could serve as a meaningful path for researchers exploring the areas of multimodality and knowledge editing.
>
> **Q.** Computational Cost
> > W1. Quality: Missing a discussion on computational overhead, which raises concerns about practical scalability.
> >
> > Q1. Computational overhead:Could you clarify the practical computational cost of MemEIC? Providing runtime comparisons or complexity analysis versus baseline methods would help.
>
> Thank you for this important question. Practical scalability is indeed crucial for real-world deployment. We provide a comprehensive computational analysis comparing MemEIC against all baselines under identical conditions (single A100 80GB, LLaVA 1.5 7B).
>
>
> **Table 1. Computational Overhead Comparison**
> | Method | Editable Params (M) | Edit Time (sec) [visual/textual] | Inference Time (sec) | Avg. Performance (CompRel, %) |
> |--------|--------------------|---------------------------------|---------------------|-------------------------------|
> | FT-LLM | 90.1 (1.28%) | 6.4 / 1.0 | 0.6 | 58.84 |
> | LoRA   | 7.7 (0.11%) | 15.6 / 5.4 | 0.7 | 61.62 |
> | MEND   | 270.3 (3.84%) | 0.04 / 0.01 | 0.6 | 2.32 |
> | SERAC  | 90.1 (1.28%) | 0.003 / 0.002 | 0.9 | 40.76 |
> | **MemEIC** | **7.7 (0.11%)** | **11.8 / 1.7** | **1.4** | **80.56** |
>
> **Key Findings**.
>
> - **Parameter Efficiency**: MemEIC requires only 7.7M editable parameters (0.11% of full model)—matching the most efficient baseline, i.e. LoRA, while achieving +18.9% better compositional reliability.
>
> - **Edit Time**: Our method outperforms naive LoRA, which must update all 16 ranks within a single module, whereas Mem-I propagates gradients through only one out of two 8-rank modules, the visual adapter for visual edits or text adapter for textual updates. This design reduces runtime, despite using the same total number of ranks.
>
> - **Inference Overhead**: MemEIC adds modest latency (1.4s vs 0.6-0.9s baselines) while delivering superior continual learning capabilities.
>
> **Detailed Overhead Analysis.**
>
> **Table 2. MemEIC Component-wise Inference Time**
> | Component | Δ Inference Time |
> |-----------|------------------|
> | Mem-E (External Memory) | 0.7s |
> | + Mem-I (Internal Memory) | +0.6s |
> | + Knowledge Connector | +0.1s |
>
> The overhead breakdown reveals optimization opportunities: Mem-E's cost is made up of query decomposition ≈ 0.5 s and retrieval ≈ 0.2 s. Query decomposition could be reduced to ~0.1s by replacing GPT-4o with efficient alternatives (FLAN-T5 Large, TinyLLaMA+LoRA).
>
> **Table 3. External Memory Cache Size Scalability Analysis**
> | Cache Size | Avg Time (ms) ±σ | Slowdown | Overhead |
> |------------|------------------|----------|----------|
> | 1 | 857.75 ± 0.15 | 1.00× | +0% |
> | 10 | 858.00 ± 0.62 | 1.01× | +1% |
> | 50 | 868.76 ± 0.92 | 1.01× | +1% |
> | 100 | 892.00 ± 0.69 | 1.04× | +4% |
> | 200 | 928.20 ± 0.94 | 1.07× | +7% |
> | 500 | 1067.56 ± 0.73 | 1.22× | +22% |
>
> We analyzed how external memory cache size affects MemEIC's retrieval time. The results show **minimal overhead growth** even as cache size increases significantly—from 1% overhead at 50 entries to only 7% at 200 entries. This growth remains manageable and is offset by the substantial performance gains in knowledge retention and compositional reasoning.
>
>
> **Conclusion.**
>
> The ultimate measure of an editing method is sustained accuracy over time. While ultra-fast methods like MEND edit in milliseconds, catastrophic forgetting renders them unreliable after multiple inferences. MemEIC achieves the optimal balance: competitive edit speed with minimal parameters while maintaining compositional reliability—essential for practical deployment where edited knowledge must persist accurately across numerous queries.
>
>
> ---
> **Q.** Robustness to Retrieval Errors
> > Q2. Robustness to retrieval errors:How does performance degrade when external memory retrieval accuracy is lower? Additional experiments or insights would strengthen confidence in real-world use.
>
> **A.** You've raised a critical concern. Real-world retrieval systems rarely achieve perfect accuracy due to noisy queries, incomplete knowledge bases, and computational constraints. Understanding MemEIC's resilience under these conditions is essential for production adoption.
>
> We evaluated robustness by simulating realistic retrieval failures, deliberately degrading accuracy to 50% - a conservative estimate for challenging deployment scenarios. Our analysis compared two key scenarios:
> - **Ideal Training, Realistic Testing**: Perfect retrieval (100%) during training, degraded retrieval (50%) during deployment
> - **Realistic Training and Testing**: Noisy conditions (50%) throughout both phases
>
> When models trained under perfect conditions face degraded retrieval during testing, performance drops substantially: 14.5% for LLaVA1.5 and 15.9% for MiniGPT-4. This represents a significant deployment risk for systems optimized under idealized laboratory conditions.
>
> However, training the knowledge connector under realistic noisy conditions from the start significantly improves robustness. This approach reduces performance degradation to 12.0% for LLaVA1.5 (2.5% improvement) and 11.4% for MiniGPT-4 (4.5% improvement), demonstrating the connector's ability to learn effective knowledge recovery strategies.
>
> From a deployment perspective, this suggests a clear best practice: train MemEIC under conditions matching expected real-world performance rather than idealized laboratory settings.
>
> These experiments (detailed in Appendix E.3.2-E.3.3) provide confidence that MemEIC maintains practical utility in real-world deployments where perfect retrieval is unrealistic.
>
> ---
>
> **Q.** Real-world qualitative results
> > W3. Significance: Evaluations primarily on synthetic benchmarks may limit real-world generalizability. The absence of user studies or real-world validation weakens claims about practicality.
>
> > Q3. Real-world generalization: Can you discuss or provide examples demonstrating MemEIC’s effectiveness beyond the synthetic benchmark scenarios? Real-world cases or qualitative results would be beneficial.
>
> > Q5. Qualitative examples:Including more qualitative examples of successful and unsuccessful compositional edits would help clearly illustrate the practical strengths and weaknesses of your approach.
>
> **A.** Thank you for raising this important point regarding real-world generalization. As described in Appendix B, CCKEB is an extended version of the VLKEB benchmark, originally constructed from the Multi-modal Knowledge Graph (MMKG) [1]. To better reflect real-world scenarios, we adopted the same construction protocol as VLKEB and utilized real-world knowledge graphs such as DB15K  and FB15K. In particular, we extracted relations from DB15K and employed maximum weight bipartite matching to select the most semantically similar object pairs. To ensure data quality and realism, all object pairs in the evaluation set were manually verified.
>
> We believe these efforts significantly improve the benchmark's alignment with real-world settings. Examples of such real-world compositional edits can be found in Table 7 (Appendix), and successful and unsuccesful model inference cases are illustrated in Figure 8.
>
> That said, we agree that providing more explicit real-world case studies and qualitative examples would further illustrate MemEIC’s effectiveness in practical settings. Due to rebuttal limitations (e.g., image upload restrictions), we are unable to include visual examples here. However, we plan to incorporate additional real-world compositional edit scenarios and corresponding outputs in the camera-ready version.
>
> ---
>
> **Q.** Simplifying Methodological Descriptions
> > W2. Clarity: Methodological sections can sometimes be overly complex; more qualitative examples would help illustrate the practical effectiveness clearly.
> >
> > Q4. Simplifying methodological descriptions:Could you offer a simplified, intuitive explanation of the dual adapters and knowledge connector? This would greatly improve clarity.
>
> **A.** We appreciate the reviewer’s suggestion, which we believe will help us further improve the clarity and accessibility of our work.
> For now, Figure 2 provides a visual overview of the Knowledge Separation (Query Decomposition and MemE) and Knowledge Integration (MemI and Connector) components. As described in Sections 1 (Introduction) and 3 (Methodology), the Dual-LoRA Adapters are inspired by the functional separation of the left and right hemispheres of the human brain: each adapter independently processes and stores visual and textual modality information. The Knowledge Connector, built on top of LoRA, functions analogously to the corpus callosum, dynamically integrating information from both modalities. Notably, it is only activated in compositional query scenarios that require cross-modal reasoning, thereby enabling effective compositional reasoning across visual and textual knowledge.
>
> To further support clarity, we plan to include pseudocode for the inference procedure in the appendix of the camera-ready version.
>
> ---
> **Limitations.**
> > The authors have not explicitly discussed limitations and potential negative societal impacts.
> >
> **A.** While we would like to clarify that we have addressed these aspects in Appendix F (Sections F.1 for limitations and F.2 for broader impacts), we acknowledge that these sections may have been insufficiently emphasized in the current version, and we will make sure to highlight them more clearly in the camera-ready version.
>
> —
>
> *References*
> - [1] Liu et al., "MMKG: Multi-modal Knowledge Graphs. In: Hitzler, P., et al. The Semantic Web". ESWC 2019

---

> > ### Author Response · Authors · 2025-08-04
> > **Edit Table**
> >
> > We apologize for the oversight. We discovered an error in Table 1 of our rebuttal (the corrected version matches the table provided in the responses to Reviewer Cssd and Reviewer qLzo).
> > The revised table is as follows:
> >
> > **Table 1. Computational Overhead Comparison**
> > | Method | Editable Params (M) | Edit Time (sec) \[visual/textual] | Inference Time (sec) | Avg. Performance (CompRel, %) |
> > | - | - | - | -| - |
> > | FT-LLM | 90.1 (1.28%) | 6.4 / 1.0 | 0.6| 58.84 |
> > | LoRA   | 7.7 (0.11%) | 15.6 / 5.4  | 0.7 | 61.62 |
> > | MEND   | 270.3 (3.84%) | 0.04 / 0.01 | 0.6 | 2.32 |
> > | SERAC  | - | **0.003 / 0.002** | 0.9 | 40.76 |
> > | MemEIC | **7.7 (0.11%)** | 11.8 / 1.7 | 1.4 | **80.56** |

---

> > > ### Author Response · Authors · 2025-08-06
> > > **Reminder**
> > >
> > > Dear Reviewer,
> > >
> > > As the author-reviewer discussion period ends on August 8th, we kindly remind you that we will be unable to communicate after that date. We would appreciate your feedback on whether our response has addressed your concerns. If you have any further questions or need additional clarification, please let us know before the discussion period concludes.
> > >
> > > Thank you for your attention.
> > >
> > > Best regards,
> > > The Authors

---

### Official Review · Reviewer_5dMc · 2025-07-03

**Clarity:** 3
**Significance:** 2
**Originality:** 3
**Rating:** 4
**Confidence:** 3

**Summary:**

The paper introduces CCKEB, a new continual and compositional editing benchmark that pairs visual relabeling with matching textual fact updates and tests models on single edits, long edit sequences, and dual-modal queries; to meet these challenges, the authors propose MemEIC, which keeps vision and text edits in separate external memories and modality-specific LoRA adapters, then selectively fuses them via a learned “Knowledge Connector” only when a compositional query requires it—resulting in near-perfect retention of early edits after hundreds of updates and over 97% reliability on compositional tasks, far surpassing both external-only and internal-only baselines.

**Questions:**

1. Could the aurthors expand your comparisons to include more recent multimodal editing methods—such as VisEdit[1] or other state-of-the-art VLM editing method—beyond MEND and SERAC? Demonstrating clear gains over these stronger baselines would substantively strengthen the paper’s claims and could help raise the score.

[1]Attribution Analysis Meets Model Editing: Advancing Knowledge Correction in Vision Language Models with VisEdit

2. Can you provide a detailed computational cost analysis of MemEIC relative to your baselines, covering training and inference time, GPU memory usage, and FLOPs for a fixed number of edits?

**Ethical Concerns:**

["NO or VERY MINOR ethics concerns only"]

**Final Justification:**

My concerns have been adequately addressed, and I have decided to increase my rating by one.

**Limitations:**

yes

**Paper Formatting Concerns:**

I have no formatting concerns.

**Quality:**

3

**Strengths And Weaknesses:**

Key strengths include:

1. CCKEB is the first to test both continual updating and compositional reasoning across vision and language.
2. MemEIC’s dual external memories, modality-specific LoRA adapters, and selective “Knowledge Connector” fusion effectively prevent cross-modal interference.
3. near-perfect retention after hundreds of edits and >97 % reliability on compositional queries across two LVLMs.

Main weaknesses are:

1. baseline method only MEND and SERAC are compared, omitting latest editing methods of VLM like visedit
2. the paper is missing a discussion of the computational cost of running MemEIC (vs other baselines), how expensive is running MemEIC?

---

> ### Author Rebuttal · Authors · 2025-07-31
>
> We thank the reviewer for recognizing our contributions and the strong performance of MemEIC. Your suggestions on baseline comparisons and computational cost analysis are valuable for strengthening our evaluation.
>
> ---
>
> **Q.  Additional SOTA Baseline Comparison**
>
> **A.** We thank the reviewer for this valuable suggestion. We share your goal of further strengthening our paper by comparing MemEIC against stronger, more recent baselines. However, existing datasets and therefore methods that support continual editing of both visual and textual knowledge are extremely limited (e.g., VLKEB, CCKEB). Additionally, some methods such as KE [1] and IKE [2] require access to editing data at test time, which reduces their practical applicability (see §4.2).
>
> **VisEdit** [3] is indeed an excellent work, and we have studied it closely. However, VisEdit focuses primarily on attribution analysis of visual representations and does not consider compositional interactions between modalities. It is designed for single-edit settings (e.g., MMEdit ), without addressing continual edit scenarios.
>
> As a stronger "lifelong" editing baseline, we therefore implemented **WISE** [4] within our CCKEB framework and evaluated it alongside FT, LoRA, MEND, SERAC, and our method on both LLaVA 1.5 7B and MiniGPT-4:
>
> **Table 1.** Average performance of CCKEB on LLaVA‑1.5
> | Method | Vis Rel | Text Rel | Comp Rel | AVG |
> |-|-|-|-|-|
> | FT(LLM) | 85.66 | 81.36 | 58.84 | 75.29 |
> | LoRA | 77.22 | 75.22 | 61.62 | 71.35 |
> | MEND | 4.22 | 4.43 | 2.32 | 3.66 |
> | SERAC | 63.17 | 86.94 | 40.76 | 63.62 |
> | WISE [4] | 81.99 | **94.16** | 48.21 | 74.79 |
> | **MemEIC (Ours)** | **98.93** | 92.48 | **80.56** | **90.66** |
>
> **Table 2.** Average performance of CCKEB on MiniGPT-4
> | Method | Vis Rel | Text Rel | Comp Rel | AVG |
> |-|-|-|-|-|
> | FT(LLM) | 76.42 | 87.23 | 41.75 | 68.47 |
> | LoRA | 77.68 | 74.39 | 58.99 | 70.35 |
> | MEND | 6.82 | 6.10 | 5.39 | 6.10 |
> | SERAC | 44.84 | 43.31 | 40.73 | 42.96 |
> | WISE [4] | 66.12 | 92.07 | 43.9 | 67.36 |
> | **MemEIC (Ours)** | **96.67** | **92.89** | **74.53** | **88.03** |
>
> > **WISE [4]** divides each FFN layer into a "main" and a "side" memory bank, explicitly separating edited knowledge from unedited knowledge to enable robust lifelong editing.
>
> > We provide detailed results for sequential gaps [0, 10, 20, 50, 100] in Tables  5 and  6 below.
>
> Despite WISE's temporal robustness (maintaining consistent performance across editing gaps), **MemEIC substantially outperforms WISE across nearly all metrics**:
> - **Visual Reliability**: MemEIC achieves +16.94% improvement on LLaVA 1.5, demonstrating superior visual knowledge editing capabilities
> - **Textual Reliability**: Results are comparable - WISE slightly outperforms on LLaVA 1.5 (+1.68%), while MemEIC shows better performance on MiniGPT-4 (+0.82%)
> - **Most critically, Compositional Reliability**: MemEIC achieves 80.56% vs WISE's 48.21% on LLaVA 1.5 - a substantial +32.35% improvement
>
> This dramatic difference in compositional reasoning reveals a fundamental limitation of current knowledge editing approaches. While recent methods like WISE [4], GRACE [5], and MEMIT [6] have made significant progress in preventing knowledge conflicts between new and old knowledge through various separation mechanisms (discrete key-value adaptors, rank-one weight modifications, memory bank separation), these approaches inadvertently **limit the interaction between edited facts** that is essential for compositional reasoning tasks.
>
> Our analysis suggests that the field's emphasis on conflict avoidance may inadvertently constrain the practical applicability of knowledge editing systems in real-world scenarios that require reasoning over interconnected knowledge. MemEIC addresses this limitation by enabling **controlled interaction** between edited knowledge while maintaining editing reliability - a capability crucial for practical deployment of knowledge editing systems.
>
> We will update the main results section (§4.1) of our paper to include WISE results. Thanks to the reviewer's suggestion to compare against a stronger SOTA baseline, we believe our paper's novelty and impact will be significantly reinforced. We will also explore extending our experimental baselines in future work to include other recent editors such as VisEdit and LTE [7].
>
> Thank you again for this excellent feedback!
>
> ---
>
> **Q. Computational Cost**
>
> **A.** Your question about computational cost analysis is particularly important for practical applications. Below, we provide a comprehensive breakdown of MemEIC's computational efficiency relative to baseline methods, evaluated under identical conditions (a single A100 80 GB, LLaVA 1.5 7B):
>
> **Table 3.** Computational Cost: Edit Parameters, Edit time, Inference time, Average Performance(in gap[0,10,20,50,100])
> | Method | Editable Params (M) | Edit Time (sec) \[visual/textual] | Inference Time (sec) | Avg. Performance (CompRel, %) |
> | - | - | - | -| - |
> | FT-LLM | 90.1 (1.28%) | 6.4 / 1.0 | 0.6| 58.84 |
> | LoRA | 7.7 (0.11%) | 15.6 / 5.4  | 0.7 | 61.62 |
> | MEND | 270.3 (3.84%) | 0.04 / 0.01 | 0.6 | 2.32 |
> | SERAC  | - | **0.003 / 0.002** | 0.9 | 40.76 |
> | MemEIC | **7.7 (0.11%)** | 11.8 / 1.7 | 1.4 | **80.56** |
>
> - **Parameter Efficiency**:  MemEIC demonstrates strong parameter efficiency, updating only 7.7M parameters (~0.11% of the model) during edit. This is enabled by a dual-component design: Mem-E uses an external memory that requires no editable parameters, while Mem-I applies two compact, brain-inspired LoRA modules to separate FFN hemispheres. This strategic design yields 80.56% compositional reliability, an 18.9% gain over the next best baseline.
>
> - **Edit Time Performance**:  Our analysis highlights modality-specific differences in editing latency. Editing latency varies by modality, with visual edits requiring more resources. MemEIC remains competitive with LoRA by propagating gradients through only one out of two 8-rank modules per modality instead of a single 16-rank module. Ablation studies show that reducing LoRA rank to 8 without modality separation leads to collapse (§4.4). Faster methods like MEND and SERAC sacrifice accuracy for speed, limiting real-world applicability.
>
> - **Inference Latency**: MemEIC introduces moderate inference latency (1.4s) versus LoRA (0.7s). External-memory methods like SERAC show intermediate latency (0.9s) but poor knowledge retention. MemEIC balances performance and speed, offering strong continual and compositional editing with practical overhead.
>
> **Table 4.** Ablation study of MemEIC's Inference time
> | Component |  Δ inference-time |
> | - | - |
> | **Mem-E** (External Memory) | 0.7 s  |
> | + **Mem-I** (Internal Memory)  | **+0.6 s** |
> | + **Knowledge Connector**  | **+0.1 s**  |
>
> - **Component-wise Overhead Analysis**: Breaking down inference time, Mem-E contributes 0.7s (0.5s for query decomposition, 0.2s for retrieval), Mem-I adds 0.6s, and the Knowledge Connector incurs minimal overhead (0.1s). Notably, the connector activates only for compositional queries, reducing average latency in practice. The decomposition bottleneck can be mitigated by replacing GPT-4o with lighter models (e.g., FLAN-T5 Large, TinyLLaMA with LoRA), potentially lowering it to ~0.1s.
>
> While computational efficiency matters, the fundamental question for any editing method remains: "How reliably and accurately does it retain edited information over time?" Methods like MEND may achieve rapid editing, but catastrophic forgetting undermines their practical utility across multiple inferences. MemEIC's framework addresses this critical limitation by combining fast editing with minimal parameter updates while ensuring robust accuracy and compositional reliability throughout extended usage.
>
> We hope this detailed computational analysis clearly addresses your concerns about MemEIC's practical viability. Your emphasis on computational practicality is essential, helping to bridge research innovations with real-world deployments.
>
> *References*
> - [1] Cao et al., "Editing factual knowledge in language models," ACL 2021.
> - [2] Zheng et al., "Can we edit factual knowledge by in-context learning?," ACL 2023.
> - [3] Chen et al., "Attribution Analysis Meets Model Editing: Advancing Knowledge Correction in Vision Language Models with VisEdit," AAAI 2025.
> - [4] Wang et al., "WISE: Rethinking the Knowledge Memory for Lifelong Model Editing of Large Language Models," NeurIPS 2024.
> - [5] Hartvigsen et al., "Aging with GRACE: Lifelong Model Editing with Discrete Key-Value Adaptors," NeurIPS 2023.
> - [6] Meng et al., "Mass Editing Memory in a Transformer," ICLR 2023.
> - [7] Jiang et al., “Learning to Edit: Aligning LLMs with Knowledge Editing”, ACL 2024
>
>
> ---
> To make it easier to follow, we have appended additional experimental results matching the table.
>
> **Table 5.** Detailed Performance of CCKEB on LLaVA‑1.5 (WISE vs Ours)
> | Method | Gap | Vis Rel | TextRel | Comp Rel |
> |-|-|-|-|-|
> | **WISE** | 0 | 81.17 | 97.69 | 49.47 |
> | | 10 | 82.06 | 96.95 | 48.10 |
> | | 20 | 81.60 | 94.70 | 47.47 |
> | | 50 | 83.09 | 93.76 | 47.83 |
> | | 100 | 82.04 | 87.72 | 48.16 |
> | **Avg** | | 81.99 | **94.16** | 48.21 |
> | | | | | |
> | **OURS** | 0 | 99.66 | 99.94 | 85.16 |
> | | 10 | 98.86 | 93.36 | 80.88 |
> | | 20 | 98.91 | 90.73 | 80.35 |
> | | 50 | 98.72 | 89.61 | 79.03 |
> | | 100 | 98.48 | 88.77 | 77.36 |
> | **Avg** | | **98.93** | 92.48 | **80.56** |
>
> **Table 6.** Detailed Performance of CCKEB on Minigpt-4 (WISE vs OURS)
> | Method | Gap | Vis Rel | Text Rel | Comp Rel |
> |-|-|-|-|-|
> | **WISE** | 0 | 81.52 | 89.27 | 44.42 |
> | | 10 | 65.71 | 91.53 | 44.22 |
> | | 20 | 61.76 | 92.06 | 43.51 |
> | | 50 | 60.94 | 93.15 | 43.77 |
> | | 100 | 60.66 | 94.35 | 43.58 |
> | **Avg** | | 66.12 | 92.07 | 43.90 |
> | | | | | |
> | **OURS** | 0 | 99.52 | 99.85 | 78.05 |
> | | 10 | 96.40 | 93.34 | 76.25 |
> | | 20 | 95.67 | 91.74 | 74.78 |
> | | 50 | 95.35 | 90.87 | 72.61 |
> | | 100 | 96.42 | 88.64 | 70.96 |
> | **Avg** | | **96.67** | **92.89** | **74.53** |

---

> > ### Author Response · Authors · 2025-08-06
> > **Reminder**
> >
> > Dear Reviewer,
> >
> > As the author-reviewer discussion period ends on August 8th, we kindly remind you that we will be unable to communicate after that date. We would appreciate your feedback on whether our response has addressed your concerns. If you have any further questions or need additional clarification, please let us know before the discussion period concludes.
> >
> > Thank you for your attention.
> >
> > Best regards,
> > The Authors

---

> ### Comment · Reviewer_5dMc · 2025-08-09
>
> My concerns have been adequately addressed, and I have decided to increase my rating by one.

---

> > ### Author Response · Authors · 2025-08-09
> >
> > Thank you very much for your feedback! We really appreciate the time you spent to help us make our submission stronger.

---

### Official Review · Reviewer_Cssd · 2025-07-06

**Clarity:** 3
**Significance:** 3
**Originality:** 3
**Rating:** 4
**Confidence:** 3

**Summary:**

This paper introduces MemEIC, a novel framework for continual and compositional knowledge editing in large vision-language models (LVLMs). The work addresses a significant limitation in existing multimodal knowledge editing approaches, which typically handle visual and textual edits in isolation and fail to support continuous, sequential updates. The authors propose a hybrid external-internal memory architecture that combines modality-specific external retrieval with dual LoRA adapters for internal memory, connected by a knowledge connector that enables cross-modal reasoning. Additionally, they introduce the CCKEB benchmark, extending VLKEB to evaluate both continual editing capabilities and compositional reasoning through a new metric called Compositional Reliability (CompRel). Experiments on LLaVA-1.5 and MiniGPT-4 demonstrate that MemEIC significantly outperforms existing methods, achieving superior performance on complex multimodal questions while effectively preserving prior edits across sequential updates.

**Questions:**

- Could you provide a detailed analysis of the computational costs (e.g., inference time, parameter overhead) introduced by MemEIC's hybrid memory architecture, including the external retrieval and knowledge connector components? How do these costs compare to simpler methods (e.g., enhanced internal memory or external retrieval alone)?
- Could you provide justification for excluding other benchmarks like MC-MKE (focused on modality consistency) or extend the experiments to broader settings to evaluate generalization?
- How does MemEIC’s hybrid approach compare to some simple baselines that mix existing internal and external memory architectures? Some discussion or results could be helpful to enhance the paper.

**Ethical Concerns:**

["NO or VERY MINOR ethics concerns only"]

**Final Justification:**

The author's response addresses my concern

**Limitations:**

yes

**Paper Formatting Concerns:**

N.A.

**Quality:**

3

**Strengths And Weaknesses:**

Strength:
- MemEIC makes a good contribution to multimodal knowledge editing by addressing limitations in single-modality and sequential updates. Its architecture, with modality separation and a dynamic knowledge connector, provides valuable insights for the field. The new framework, benchmark, and metric are good contributions.
- Comprehensive experiments, including ablation studies, demonstrate consistent performance gains across backbone models (e.g., LLaVA-1.5, MiniGPT-4) and evaluation metrics, particularly excelling in compositional reasoning tasks.

Weakness:
- Computational Overhead: The paper doesn’t sufficiently address the increased complexity introduced by the hybrid memory architecture. A runtime analysis would clarify whether the performance justifies the additional computational cost compared to simpler alternatives.
- Limited Benchmarks: The reliance on the CCKEB benchmark raises concerns about generalizability. Validation on new, diverse benchmarks (e.g., MC-MKE) would strengthen the experimental scope.
    - MC-MKE: A Fine-Grained Multimodal Knowledge Editing Benchmark Emphasizing Modality Consistency (CVPR 2025)
- Comparison Fairness: MemEIC combines internal and external memory techniques, making it unclear whether the observed gains are due to the novel design itself or the increased resource usage. A clearer analysis comparing hybrid models to standalone internal/external methods would enhance credibility.

---

> ### Author Rebuttal · Authors · 2025-07-31
>
> We thank the reviewer for their time and thoughtful evaluation of our work. Below, we address the specific comments in detail, and we would be happy to further engage in discussion if needed.
>
> ---
>
> **Q. Computational Cost**
> > **W1.** Computational Overhead: The paper doesn’t sufficiently address the increased complexity introduced by the hybrid memory architecture. A runtime analysis would clarify whether the performance justifies the additional computational cost compared to simpler alternatives.
>
> > **Q1.** Could you provide a detailed analysis of the computational costs (e.g., inference time, parameter overhead) introduced by MemEIC's hybrid memory architecture, including the external retrieval and knowledge connector components? How do these costs compare to simpler methods (e.g., enhanced internal memory or external retrieval alone)?
>
> **A.** Thank you for raising a great question. While our approach demonstrates strong editing capabilities at the research level, ensuring practicality in real-world applications requires that we keep computational costs under control. To that end, we evaluated the computational overhead of our method versus those of the baselines under identical conditions (a single A100 80 GB, LLaVA 1.5 7B):
> \
> \
> **Table.** Computational Cost: Editable Parameters, Edit time, Inference time, Average Performance(across gap[0,10,20,50,100])
> | Method | Editable Params (M) | Edit Time (sec) \[visual/textual] | Inference Time (sec) | Avg. Performance (CompRel, %) |
> | - | - | - | -| - |
> | FT-LLM | 90.1 (1.28%) | 6.4 / 1.0 | 0.6| 58.84 |
> | LoRA   | 7.7 (0.11%) | 15.6 / 5.4  | 0.7 | 61.62 |
> | MEND   | 270.3 (3.84%) | 0.04 / 0.01 | 0.6 | 2.32 |
> | SERAC  | - | **0.003 / 0.002** | 0.9 | 40.76 |
> | MemEIC | **7.7 (0.11%)** | 11.8 / 1.7 | 1.4 | **80.56** |
>
> - **Editable Parameters**:  Our method requires significantly fewer editable parameters than most methods — 7.7 M (≈ 0.11 % of the full model). Mem-E incurs no extra editable parameters at all, since it simply stacks new knowledge in external memory. Mem-I restricts updates to two small, brain-inspired LoRA modules (one per FFN “hemisphere”), so it only ever edits those lightweight parameters. Despite this extreme parameter efficiency, Mem-I achieves outstanding retention: Mem-I exhibits exceptional retention—maintaining performance even as the interval between updates grows (Gap: 0, 10, 20, 50, 100 )—and achieves an average Compositional Reliability of 80.56%: an 18.9% improvement over the runner-up method.
> - **Edit Time** :  We measured editing latency separately for visual and textual knowledge. As expected, editing visual tokens takes longer than editing text tokens—visual embeddings are simply heavier. Nevertheless, our method still runs faster than LoRA, the previous state of the art. Naive LoRA must update all 16 ranks within a single module, whereas Mem-I propagates gradients through only one out of two 8-rank modules, the visual adapter for visual edits or text adapter for textual updates. This design reduces runtime, despite using the same total number of ranks. Importantly, our ablation study (§4.4) shows that naively reducing LoRA’s rank to 8 in an attempt to reduce edit time causes representation collapse and significantly degrades editing accuracy. In contrast, Mem-I preserves representation quality while cutting edit time. Other fast editors (e.g., MEND, SERAC) trade speed for severely degraded compositional reasoning—unacceptable for real use.
> - **Inference Time** :  Among internal-memory approaches, LoRA remains fastest at around 0.7 s per query. External-memory methods like SERAC are slower (≈ 0.9 s) due to classifier overhead and query matching, and they underperform at knowledge internalization. Our model strikes a trade-off between latency and performance: it adds only modest latency while delivering far superior continual and compositional editing gains.
> - **Ablation of Inference Overhead** :  As requested in Q1, we also isolated the overhead of each Mem-EIC component during inference. Starting from Mem-E alone, query decomposition costs ≈ 0.5 s and retrieval ≈ 0.2 s. Adding Mem-I increases latency by 0.6 s, and the Knowledge Connector adds another 0.1 s. Note that Mem-E’s retrieval is lightweight (0.2 s); the bulk of its cost is query decomposition, which could be greatly reduced by replacing GPT-4o with a smaller model (e.g. FLAN-T5 Large or TinyLLaMA with LoRA)—we estimate 0.5 s could drop to 0.1 s. Moreover, the Knowledge Connector only activates on compositional queries, so average latency in practice will be lower.
>
>  **Table.** Ablation study of MemEIC's Inference time
> | Component |  Δ inference-time |
> | - | - |
> | **Mem-E** (External Memory) | 0.7 s |
> | + **Mem-I** (Internal Memory)    | **+0.6 s** |
> | + **Knowledge Connector** | **+0.1 s** |
>
>
> In conclusion, computational cost is important, but the ultimate measure of an editing method is **'how long and how accurately it retains the edited information'**. Even if a method like MEND edits very quickly, catastrophic forgetting renders its outputs unreliable across multiple inferences. Our Mem-EIC framework, by contrast, achieves fast edits with minimal parameter updates while preserving accuracy and compositional reliability over time.
>
> ---
>
> **Q. Other Benchmarks**
> > **W2.** Limited Benchmarks: The reliance on the CCKEB benchmark raises concerns about generalizability. Validation on new, diverse benchmarks (e.g., MC-MKE) would strengthen the experimental scope.
>
> > **Q2.** Could you provide justification for excluding other benchmarks like MC-MKE (focused on modality consistency) or extend the experiments to broader settings to evaluate generalization?
>
> **A.** Reviewer **Cssd**, thank you for pointing us to MC-MKE. That work appears in the ACL Findings track, which is currently underway, so its underlying data have understandably not yet been released. Even if those data was available, MC-MKE performs only a single edit on one modality—either textual or visual knowledge—and evaluates it immediately, whereas our benchmark continuously updates knowledge over time, deliberately inserting gaps between edits and evaluation to test true lifelong editing capability. Furthermore, unlike MC-MKE, we alternate between textual and visual edits and introduce a new metric, evaluation protocol, and dataset to assess whether combinations of those edits remain coherent. To the best of our knowledge, this is the first benchmark for continual, compositional knowledge editing.
>
> ---
>
> **Q. MemEIC vs Baselines**
> > **W3.** Comparison Fairness: MemEIC combines internal and external memory techniques, making it unclear whether the observed gains are due to the novel design itself or the increased resource usage. A clearer analysis comparing hybrid models to standalone internal/external methods would enhance credibility.
>
> > **Q3.** How does MemEIC’s hybrid approach compare to some simple baselines that mix existing internal and external memory architectures? Some discussion or results could be helpful to enhance the paper.
>
> **A.** We thank the reviewer for raising the important point of how MemEIC’s hybrid external-internal approach compares to simpler combinations of internal and external memory.
>
> First, we would like to clarify SERAC’s role in this landscape. Although SERAC stores edit samples in an edit memory and is thus classified as an external-memory method, during training it also learns a separate counterfactual model in addition to the base model, updating additional parameters in the process. Despite this hybrid-style mechanism, our experiments in §4.2 (Table 1) show that MemEIC outperforms SERAC across all metrics.
>
>
> Crucially, this advantage is not due to simply adopting a hybrid memory structure. Rather, it comes from systematically improving each component(external and internal) in isolation. Our ablation studies demonstrate that both Mem-E (external) and Mem-I (internal) modules individually surpass naive editing baselines in terms of accuracy.
>
> To make these structural benefits explicit:
> - Visual Editing (Mem-E, §4.3). By more effectively leveraging rich visual context during retrieval, our enhanced Mem-E module achieves substantially higher visual editing accuracy than previous external-memory methods.
> - Compositional Reliability (Knowledge Connector, §4.5). Our self-attention-based Knowledge Connector plays a pivotal role in reconciling information from internal parameters and external retrieval. This mechanism is especially beneficial for compositional editing scenarios, driving the notable gains we observe on our Compositional Reliability metric.
>
> We appreciate the reviewer’s suggestion and will add an explicit discussion of these distinctions in the revised manuscript.

---

> > ### Comment · Reviewer_Cssd · 2025-08-06
> >
> > The author addresses my concern. These rebuttal updates shall be included in the main paper.

---

> > > ### Author Response · Authors · 2025-08-06
> > >
> > > Thank you for confirming that our rebuttal addressed your concerns. We will ensure that the suggested updates are fully integrated into the final manuscript.

---

### Note · Authors · 2025-08-12

We thank the AC and reviewers. During the rebuttal, key concerns were resolved; **5dMc raised their rating**, and others confirmed our updates addressed their issues.

**1. Practicality (cost vs. utility)** *(Cssd, 5dMc, qLzo)*
MemEIC edits only **7.7M** params (**0.11%**), with a 11.8s/1.7s (visual/text) edit latency  and 1.4s inference time.
Despite the tiny footprint, Compositional Reliability was increased by 18.9% over the next best baseline.

**2. Stronger baselines** *(5dMc, Cssd)*
We added a comparison to the strong lifelong editor baseline **WISE (NeurIPS’24)**. On LLaVA‑1.5, MemEIC(Ours) vs. WISE: Visual Reliability +16.94 (98.93 vs. 81.99) and CompRel +32.35 (80.56 vs. 48.21).

**3. Robustness of noisy retrieval** *(qLzo)*
Some reviewers were concerned how errors from incorrect retrievals in Mem-E would propagate through Mem-I and affect the final result. As described in App. E.3.2 (Fig. 7), noise-aware training of the Knowledge Connector improves **noisy-setting reliability by 2.5%**, supporting real-world resilience.

**4. Scale-up · Statistics** *(3Fzx)*
- New ≥13B results: On LLaVA-1.5-13B, MemEIC reaches 76.62 CompRel compared to 61.67 (LoRA) and 56.80 (FT), confirming scalability without redesign.
- We added std/paired t-tests (p<0.05) showing Dual-LoRA’s effectiveness at preventing representation collapse.

**5. Generalization** *(Cssd, qLzo, 3Fzx)*
- Across backbones: Large CompRel gains on both LLaVA-1.5 and MiniGPT-4.
- Data realism: CCKEB augments VLKEB with MMKG-derived edits, and provides manually verified eval pairs, guaranteeing realistic fact changes.
- Unavailability of other benchmarks: MC-MKE (ACL-Findings’25) is single-edit/unimodal (no continual+compositional editing) and not yet public.
- Applicability to Visual-only tasks: The framework is modular - in unimodal settings the Connector remains inactive and Mem-E/Mem-I operate unchanged.

**6. Outcome of the discussion**
   - **Cssd**:  “The author addresses my concern”
   - **5dMc**: "My concerns have been adequately addressed"; **rating increased**
   - **3Fzx**: “ You have addressed all my concerns.”

MemEIC is the first to target **continual + compositional** multimodal editing with a matching benchmark/metric, while being **practically viable** (tiny footprint, low latency), **strong** (vs. WISE and others), **robust** (to noisy retrieval and long sequences), and now **validated to scale** (≥13B). We will integrate all additions in the camera-ready.

---

### Decision · Program_Chairs · 2025-09-17

**Decision:**

Accept (poster)

**Comment:**

The paper proposes MemEIC, a framework for continual and compositional knowledge editing in LVLMs. It integrates external memory retrieval and modality-specific LoRA adapters with a brain-inspired “knowledge connector” that selectively fuses cross-modal knowledge. Also the authors introduce the CCKEB benchmark and a new metric to evaluate both continual and compositional editing.

The strengths highlighted by reviewers include the novelty of targeting multimodal continual editing, the hybrid architecture that avoids cross-modal interference, and the benchmark/metric contribution that provides tools for future research. Empirical results are strong and supported by ablations, while the paper is generally well-structured and clear.

The weaknesses noted include limited evaluation breadth (primarily on CCKEB, without stronger baselines like VisEdit or other recent editing methods), insufficient discussion of computational cost and scalability, and some ambiguity in claims and result reporting. Concerns about generalizability to real-world or larger LVLMs remain.

During the rebuttal, authors clarified details of methodology, addressed dataset and comparison questions, which reassured some reviewers though not all concerns were fully resolved. Overall, I believe the paper is timely, makes a contribution to the field.